# How high-intensity sensory consumption fills up resource scarcity: The boundary condition of self-acceptance

**Liangjun Peng**[1], **Yuxin Peng**[2], **Haiyan Luo**[3]*, **Yeying Deng**[4]

**1** School of Economics and Management, Chongqing Jiaotong University, Chongqing, China, **2** College of Life Sciences, Henan Normal University, Xinxiang, Henan, China, **3** School of Education, Universiti Teknologi Malaysia, Johor Bahru, Malaysia, **4** Faculty of Economics and Management, Universiti Kebangsaan Malaysia, Bangi Selangor, Malaysia

* luohy@gzgs.edu.cn

**Data Availability Statement:** All relevant data are within the manuscript and its Supporting information files.

**Funding:** The author(s) received no specific funding for this work.

## Abstract

### Objective

Everyone in life will experience resource scarcity, which causes self-discrepancy. It is widely known that individuals participate in reactive consumption to solve the problems of self-discrepancy and resources scarcity. This kind of consumption may be symbolically related to the essence of the resource scarcity or may occur in an unrelated domain. This study proposes a theory for "filling up" one's resource scarcity through high-intensity sensory consumption (HISC).

### Methods

We used different methods, including one-way analysis of variance (ANOVA), linear regression, mediating effect, and moderating effect, to test the four hypotheses. Four experiments in the study were conducted from May 2022 and August 2022 and involved undergraduates from a university and volunteers recruited online. All participants are adults and verbally agree to participate voluntarily. Study 1a (N = 96 (male 47, female 49), participants from a business school in China) measured resource scarcity in the laboratory experiments and verified the effect of resource scarcity on consumer HISC preference by using linear regression (H1). Study 1b (N = 191 (male 98, female 93), students and teachers from a university in China) measured resource scarcity in the laboratory experiments and manipulated positively and negatively valenced experiences. Using the PROCESS SPSS Mode l, we verified that negatively valenced stimuli also lead to higher levels of arousal, which in turn restores the self-discrepancy caused by resource scarcity (H2). Study 2 (an online experiment, N = 182 (male 91, female 91), participants from China) manipulated the resource scarcity in a color sensory stimulant context, replicating the preliminary effect and examined the mediating effect of the self-worth by using the PROCESS SPSS Mode 4 (H3). Study 3 (an online experiment, N = 251 (male 125, female 126), participants from China) manipulated resource scarcity and self-acceptance in the tactile sensory experience, and tested the moderating effect of self-acceptance by using the PROCESS SPSS Mode 8 (H4).

**Competing interests:** The authors have declared that no competing interests exist.

## Results

Four studies suggest that not only do individuals facing resources scarcity prefer HISC but also that this consumption is mediated and moderated by self-worth and self-acceptance, respectively. This preference for HISC is negated when individuals have high self-acceptance traits. The findings are tested in the auditory domain (as evidenced by a propensity for louder volume), the visual domain (as evidenced by a propensity for more intense colors), and the tactile domain (as evidenced by a propensity for more intense need for touch). The findings also demonstrate that individual preferences for HISC is shown to operate regardless of the valence (positive valence vs. negative valence) of the sensory consumption.

## Conclusions

Across four experiments, we find that individuals who are subjected to resource scarcity show a preference for high-intensity sensory consumption in the auditory, visual, and tactile domains. We also find that both negatively and positively valenced sensory stimuli have the same impact on resource-scarce individuals' preference for HISC. Furthermore, we demonstrate that the sense of self-worth significantly mediates the effect of resource scarcity on HISC. Finally, we reveal that self-acceptance moderates the effect of resource scarcity on HISC preference.

## 1. Introduction

Everyone in life will have the experience of resources scarcity. For example, people who lose weight control their diet and always perceive the scarcity of calories; poor people are concerned about their livelihood and always perceive the scarcity of money; busy people have too much to do and always perceive the scarcity of time. Science focuses on the topic of "scarcity" in a paper titled " Some consequences of having too little" [1] and further explores the consequences of resource scarcity in "Poverty hinders cognitive function." More attention is paid to how resource scarcity impacts the interaction between consumers and the environment in the field of marketing [2]. Products are symbolic, always regarded as an extension of the self [3], and the practical utility they offer go beyond the mere functionality. Products in fact also play a significant role in self-restoration after individuals subjected to resource scarcity. While the relationship between resource scarcity and the counter-hedonic consumption and selfishness has already been examined, the effect of resource scarcity on sensory consumption (lacking branding and other contextual cues) remains unexamined. Sensory consumption has been defined as a consumption experience that is processed by one or more of the five sensory modalities of vision, sound, smell, taste, and touch [4], and recent research in this area has mainly focused on sensory marketing, that involves the senses to influence consumer judgment, perception, and behavior [5]. Every year, thousands of brands around the world can't wait to introduce themselves to consumers. In the consumption context, many brands attract consumers through one or more of the five sensory stimuli [6–8]. In this study, we propose that resource scarcity may also affect sensory consumption of consumers. We examine whether consumers experiencing resource scarcity have a propensity to engage in high-intensity sensory consumption across different sensory domains and whether such consumption occurs through the mediation of self-worth. In doing so, we propose the new proposition that

"filling up" one's resource scarcity through HISC has the impact of restoring self-discrepancy and may be a unique compensatory consumption form when experiencing resource scarcity.

Individual often experience reminders of various resource scarcity, which can alternate their subsequent cognition, emotion, and decision making [9]. The extensive literature suggests that a resource scarcity mindset alters consumer decision-making [10], and individuals respond to resource scarcity by engaging in reactive consumption [11, 12]. Most of those studies investigates reactive consumption that is symbolically associated with the nature of resource scarcity, often dubbed as 'within-domain' compensation [13]. For instance, as descriptions in the literature on conspicuous consumption [14], individuals with a scarcity for social status choose brands and products that obviously indicate their socioeconomic status to others, such as prominent luxury brand logos [15] (Han et al. 2010). Analogously, consumers who feel resource scarcity have also been shown to increase consumption of status or positional products under conditions of economic hardship or financial dissatisfaction [16, 17] and consumption of highly caloric foods [18, 19]. Past studies have mainly described "within-domain" compensation as self-restoration due to the typically symbolic public property of consumption; the consumption may be a signaling mechanism. A more limited body of research has studied reactive consumption in the domains unrelated to resource scarcity [20, 21], which some have referred to as 'across-domain' compensatory consumption [22]. Although studying psychological pathways such as scarcity reduction and control restoration [23], these studies have not investigated the effect of resource scarcity on such 'across-domain' consumption of high-intensity senses.

In the present research, we examine whether individuals experiencing resource scarcity are inclined toward experiences and products that are more intense at the basic sensory level (such as more visually intense stimuli with status-enhancing or symbolic cues). Furthermore, we investigate whether individuals experiencing resource scarcity tend to have low self-worth, and whether this tendency exhibits a propensity for high-intensity consumption along sensory domains other than the visual, such as auditory. We also research the underlying mechanisms that guide the observed effects. Batra [24] defined high-intensity sensory consumption as a kind of compensatory consumption that greatly stimulates the senses through enhancing the intensity of key sensory attributes, which occurs by increasing levels of arousal to restore self-worth. Numerous extant studies on sensory stimulation show that key attributes such as saturation and hue levels of colors and higher decibel levels of sounds lead to higher arousal levels [25–28]. We propose that individuals seek high-intensity senses after experiencing resource scarcity for the restoration of self-worth, and that elevated arousal levels minimize rumination [13] on thoughts related to resource scarcity in this process. Although previous studies have examined different outcomes related to resource scarcity [29], we not only investigate a unique compensatory consumption in response to resource scarcity, but also demonstrate an underlying mechanism by which this compensatory consumption can serve as the way to restore self-worth.

In the next section, the theoretical framework and hypotheses are developed, based on the literature on resource scarcity, self-worth, compensatory consumption, and self-acceptance. Four experiments are presented to check our hypotheses, and we find support for our theory of filling up resource scarcity via high-intensity sensory consumption. We close with a general discussion of our contribution and substantive implications, in addition to the limitations and future research.

## 2. Literature review, conceptual development, and hypotheses

### 2.1 Resource scarcity and high-intensity sensory consumption

Resource scarcity is a universal aspect of human life and a subjective feeling that there are fewer resources available than needed [9, 30]. Gong and Goldsmith et al. [31] defined resource

scarcity as perceived or observed discrepancy between one's current available resources and a higher, more desirable resources such as money, time, food, or other kinds of consumable and measurable resources to meet vital needs. Different resource scarcity has different representations of the sense of lacking, some are mindset, others are psychological [1]. Although this sense of lacking arises from different kinds of resources scarcity, they have one critical common characteristic: an adverse discrepancy [32] (Goldsmith et al. 2021). Resources scarcity is essentially a subjective feeling, which is activated by two factors: objective insufficiency and subjective insufficiency [33]. People with scarce resources will have negative emotions such as deficiency and unhappiness if it is difficult to directly address resource scarcity in a short period of time [23]. Sharma's research shows that the unpleasant and inferiority affect associated with financial deprivation motivates people to choose, attend to, and consume scarce products rather than comparable abundant goods. Research has shown that individuals employ various mechanisms to defend their unhappiness and depression in the face of resource scarcity [20, 34, 35]. Following the experience of resource scarcity, people are motivated to make up for these deficiency and unhappiness through many different defense mechanisms. Two special self-restoration mechanisms that have gained sizeable research interest are compensatory consumption and self-acceptance.

A first set of responses to resource scarcity includes participating in compensatory behavior to mitigate the negative emotions of scarcity [11, 23, 36, 37]. Those compensatory consumption can be symbolically associated with the original source of the resource scarcity [16, 31] or can occur in domains outside of the resource scarcity [23, 37]. For instance, Griskevicius et al. [16] (2013) find that cues of resource scarcity led people from lower socioeconomic status childhoods to prefer consumption of riskier rewards. Similarly, Gong et al. [31] find that reminders of resource scarcity pose a threat to consumers' personal freedoms, which in turn leads them to choose brands or products supported by a minority in an attempt to reassert their diminished freedom. In this context, a large body of literature has documented the relationship between characteristics of consumer demand (such as status needs) and conspicuous consumption as a compensatory mechanism. For example, researchers [15, 36, 38] find that consumers with high need for status or power prefer luxury goods that are conspicuously branded because these products symbolize status or power [3]. Lisjak et al. [13] call the consumption of products that symbolize accomplishment in the domain of resource scarcity as "within-domain" compensation. They argue that such within-domain compensation behavior can serve as a psychological salve to restore the self-discrepancy triggered by resource scarcity.

Although a rich body of research focuses on within-domain compensation consumption to offset a resource scarcity, all compensatory consumption need not be in the same domain as self-discrepancy. This compensatory consumption, which is not related to the source of the resource scarcity, is known as 'across-domain' compensatory consumption [13], although other names for this type of compensatory consumption include 'fluid compensation' [35] and 'cross-domain compensation' [39]. For example, researchers have shown that distinctiveness of the prime increases the willingness of the consumer to pay for their favorite foods and that thirst, hunger, or sexual arousal increases the option of unique products [39]. These findings suggest that consumption outcomes are also known to be not always relevant to the primed need. For example, consumers who are in poverty consume more high-calorie food intake [40, 41], and low income leads to increased consumption of higher-status or positional products [42, 43]. Therefore, in the absence of a chance to symbolically fulfill the self through scarcity-related consumption, compensatory consumption in an unrelated domain may alleviate the threat of resource scarcity. Kim S. and Derek D. [20] suggested that distractive experiences, which are unrelated to the property of the self-threat, such as listening to music or the consumption of cookies, can be used to mitigate self-awareness as a reaction to some kind of

threat. Similarly, when financial scarcity poses a threat to consumers' freedom of choice, the distractive experience of seeking diversity can help reduce that threat [44]. They find that participants who had experienced resource scarcity preferred more distracting activities.

Similarly, relevant to present research are other outcomes of resource scarcity, including sensory or perceptual biases deriving from the experience of resource scarcity. For instance, research on affect-gating suggests that an organism's emotional state can make it more sensitive to different types of sensory inputs, such as tactile or visual stimulation [45]. Similarly, food scarcity leads to a 32% expansion of adjustment in mammalian orienting, as it amplified variability in visual sensory threshold responses [46]. Using ethnographical methods, Scott et al. [47] show that consuming high-intensity sensory experiences, such as electric shocks and freezing water, can help create a sense of a fulfilled life by providing an opportunity to escape from the mundane self. Through eight studies, Yang and Zhang [48] show that resource scarcity can affect consumers' preferences for high-stimulation sensory consumption, such as haunted houses and horror movies. It is reasonable that high intensity sensory consumption as a kind of compensatory consumption is used to restore the self-discrepancy caused by resource scarcity. Extrapolating from such studies the effects of resource scarcity on increased sensitivity to sensory input, increased taste indulgence, and perceptual bias, we propose that resource scarcity can trigger a general preference for more intense sensory consumption. We hypothesize that consumers facing resource scarcity will prefer high-intensity sensory consumption and that this high-intensity sensory consumption will extend across all sensory domains.

**H1:** Consumers subjected to resource scarcity will exhibit a propensity for HISC compared to those who are not subjected to resource scarcity.

## 2.2 Negatively valenced HISC

Some previous studies indicate that positively and negatively valenced arousing stimuli may have a different effect on self-restoration. As suggested by Gorn et al. [49], compared to low-arousal states, high-arousal states reduce cognitive processing capability, leading to a polarization of judgments toward an individual's existing emotional state, making positive or negative affect more obvious. They find that the polarizing effect is more pronounced for self-referent claims. Shen and Wyer [50] report the influencing factors of positively and negatively valenced product arousing information and find that the influence of negatively valenced arousing information is relatively greater. Alberts and Thewissen [51] explores that mindfulness arousal has a significant effect on memory of positively and negatively valenced stimuli. They find that participants who received the brief mindfulness arousal have significantly lower memory of negatively valenced stimuli. Kaneko et al. [52] recently introduced the EmojiGrid as valid and language-independent self-assessment tools to measure food-related emotional arousal. The results show that the relationship between valence and arousal is a well-known U-shape.

The question then is raised as to whether the HISC must be positively valenced for resulting in self-restoration. Research on the competing theory of arousal bias [53] suggests that positive and negative sources of arousal have similar impacts on attentional biases, and the amygdala is activated in both cases [54–56]. This stream of research shows that high-intensity arousal is more important than valence in the aspect of biasing attention or recall of target-related stimuli or information. As early as 2001, Garavan [57] find that Amygdala activation, relative to low arousal, is significantly increased for both negatively and positively valenced stimuli and does not differ for the two valences. Further research by Baeken et al. [58] later find that the negatively and positively valenced stimuli are selected to be equal in arousal levels, because emotional valence and arousal may be controlled by different nervous systems. In fact, Mehta

et al. [7] even find that louder (at higher decibel levels) negative ambient noise increases arousal levels more than negative sounds at lower decibel levels. Hill et al. [59] find that both negatively and positively valenced stimuli can facilitate mirror neuron activity, which provides empirical support for a bidirectional link between the mirror neuron system and emotion. Tanck et al. [60] examine gender differences in affective and evaluative responses to body checking. They find that both negatively valenced body checking and positively valenced body checking led to increased negative affect in both men and women.

Prior literature suggests that people experiencing different kinds of scarcity may be interested in enhancing their arousal to restore self-worth through negative valenced stimuli. Griskevicius et al. [16] find that people experiencing economic deprivation were more likely to seek entertainment with negatively valenced stimuli and suggest that this preference may be a way for people to increase their arousal levels and temporarily escape scarce-related stress and anxiety in order to restore their self-worth. Mogilner et al. [61] find that people experiencing time scarcity are more inclined to choose products or experiences that offer instant gratification and excitement, even if those experiences are negative in nature. This preference for negatively valenced stimuli, they explain, may be driven by a desire for arousal in the face of limited resources. Meier et al. [62] find that people who experience food scarcity prefer to view vague stimuli as threats and show higher physiological responses to negatively valenced stimuli. They suggest that this increased sensitivity to negatively valenced stimuli may be a way people respond to potential threats. These studies suggest that people experiencing scarcity may be more interested in negatively valenced stimuli to increase their arousal levels and cope with the stress and anxiety associated with scarcity to restore their self-worth. Consequently, we hypothesize that both positive and negative valenced HISC will increase the arousal levels, thus filling the psychological emptiness resulting from resource scarcity and gaining a kind of psychological compensation. In other words, the process of propensity for HISC by resource-scarce consumers will be independent of the valence of the sensory consumption, since we expect both to be compensatory consumption for resource-scarce consumers.

**H2:** Consumers subjected to resource scarcity who engage in negatively valenced HISC will present increased propensity levels similar to those who engage in positively valenced HISC.

## 2.3 Resource scarcity, self-worth, and mechanisms of self-restoration

Self-worth is a favorable opinion or estimate of one's self, which depends on one or more areas of one's identity. Specifically, the contingency of self-worth is the category or domain of outcomes in which an individual stakes their self-esteem, so that an individual's perception of her or his worth depends on perceived successes, failures or adherence to self-standards in that category or domain [63]. Self-worth, as defined by Hibbert [64], refers to the person's level of value and acceptance of being and abilities. To help understand this concept, Shah et al. [1] proposed a state-dependent explanation of scarcity: they posit that no matter how limited the resource itself is (time, money, food, etc.), mere feeling of not having enough can trigger a scarcity mindset. Self-worth is based on an individual's perception and self-evaluation of social objects [65, 66]. Therefore, self-worth is largely determined by an individual's perceptions of the outside world, influenced by both positive and negative events [67]. For example, an individual's position in the social class hierarchy leads to different psychological characteristics [68]. People with scarce resources tend to be at the bottom of the social class hierarchy, while those with abundant resources tend to be at the top of the social class hierarchy [69]. Such discrepancies can have wide-ranging effects on the physiology, psychology, and behavior of

different social class hierarchy [70, 71]. Individuals who are limited by social resources and status for a long time are prone to be influenced by external situational factors in their psychology and behavior [68] (Kraus et al., 2012), and have a lower self-worth and a higher threat sensitivity [72–74]. Childhood experiences of resource scarcity affect self-confidence and self-esteem even in adult consumers [75, 76]. On the contrary, individuals with more social resources tend to be more respected and rewarded. With low variability, individuals interpret resource scarcity as a psychological threat to individuals.

Following such psychological threat, individuals will be motivated to take action through various defense mechanisms to restore self-worth. Self-worth, as a self-perception, is a source of psychological vulnerability [77]. Therefore, resource scarcity can produce an aversive psychological state that threatens the contingency of self-worth (Steele 1988), induces self-discrepancy, which can result in a state of discomfort [13, 78]. Cannon et al. [23] propose that consumers with scarce resources will have negative emotions such as unhappiness and disgust if it is difficult to solve the root causes of scarcity in a short period of time. According to self-worth theory, consumers will take countermeasures when their self-worth is threatened to mitigate or avoid the negative impact of threats on their self-worth [79]. For example, consumers will mitigate the negative emotions caused by resource scarcity through compensatory consumption, which is also a process of emotion regulation. The high intensity of sensory consumption, as a kind of compensatory consumption, can arouse people's emotions. Arousal, defined as a state of motivation, can motivate an individual from continuous drowsiness, in which one feels soporific, to extreme energized, in which one feels alertness [80–82]. Arousal levels correlate with emotional intensity [83–85]. Previous studies have found that arousal may arise from three types of sensory stimulus attributes [86], including psychophysical attributes (such as warmth, loudness, and brightness), physiologically relevant ecological attributes (such as sex and fear), and collative attributes (such as novelty of stimuli). Present study involves the first type of arousal. Prior research shows that different intensities or levels of sensory stimuli cause varying levels of arousal. For instance, brighter colors with high saturation are more arousing than dull colors with low saturation, and certain hues of color, such as orange, are more arousing than other hues, such as blue [26]. Therefore, past scholars have often used bright background colors experimentally to induce arousal [87, 88]. In the domain of auditory research, Mehta et al. [7] find that louder sounds with higher decibel levels lead to higher arousal levels than quieter sounds with lower decibel levels. In different sensory domains, it appears that more intense stimuli, such as brighter colors and louder sounds, are related with higher levels of arousal. We expect consumers subjected to resource scarcity to have a lower sense of self-worth. To mitigate the negative impact of resource scarcity, consumers will restore self-worth through high-intensity sensory consumption. In other words, consumers experiencing resource scarcity lead to a propensity for high-intensity sensory consumption, which is mediated by self-worth.

**H3:** The impact of resource scarcity on high-intensity sensory consumption will be mediated by self-worth.

## 2.4 Self-acceptance

As noted above, one mechanism that has been shown to restore overall self-worth is self-acceptance [89] (MacInnes 2006). Self-acceptance refers to the unconditional acceptance of oneself, regardless of one's abilities and performance, and whether one achieves the desired state [89, 90]. Strengthening an individual's belief in self-acceptance is viewed as a cornerstone of mental health and psychological well-being [91]. Individuals with Self-accepting also exhibit a

separation of self-worth from whether they meet ideal standards, which can enhance the individual's tolerance for setbacks and mitigate the threat of self-discrepancy [92]. In addition, self-acceptance also has a positive effect on the subsequent behavior of individuals when faced with self-discrepancy. Individuals with low self-acceptance associate information about self-discrepancy with self-worth, and therefore engage in self-restoration actions such as compensatory consumption to mitigate discrepancy. On the contrary, individuals with high levels of self-acceptance strip this negative information from self-worth, seeing it not as a source of discrepancy or threat, and without the need for self-restoration [93].

Self-worth and self-acceptance are two related but distinct concepts in psychology that play different roles in regulating our thoughts, emotions, and behaviour [94]. Self-worth refers to our overall sense of value and worth as a person, while self-acceptance refers to our ability to acknowledge and accept all parts of ourselves, including our flaws and weaknesses [95]. Self-worth can be seen as a mediator in the sense that it can influence how we feel about ourselves and how we behave in response to different situations [77]. For example, if someone has high self-worth, they may be more confident in their abilities and less likely to experience negative emotions such as anxiety or self-doubt. Self-acceptance, on the other hand, can be seen as a moderator in the sense that it can influence the relationship between other factors and our sense of self-worth [96]. For example, if someone has a high level of self-acceptance, they may be less likely to be affected by negative feedback or criticism from others, which can help to protect their self-worth. On the contrary, if someone has low self-acceptance, they may be more vulnerable to negative feedback and criticism, which can undermine their sense of self-worth. In short, although self-worth and self-acceptance are closely related, these different processes still have independent room for operation, which highlights the importance of understanding these two concepts to influence consumer behaviour differences [96].

In this study, we introduce self-acceptance as a moderator of individuals' response to self-discrepancy. By severing the link between one's own assessment of attributes (eg, skills, abilities) and self-worth, self-acceptance naturally leads to an evaluation of self-discrepancy as benign to one's delf-worth. Therefore, individuals who are high on self-acceptance are less likely to engage in compensatory consumption to restore self-discrepancy. For example, research by Kim and Gal [93] find that individuals with high self-acceptance engage in adaptive rather than compensatory consumption in response to self-deficit information. According to Smith and Petty [97], individuals with high self-acceptance have more confidence in their abilities and the achievement of their efforts regardless of resource abundance, which can buffer the impact of adverse events, while individuals with low self-acceptance lack confidence in their abilities and the achievement of their efforts when they are subjected to resource scarcity. Research shows that self-acceptance is positively correlated with self-worth [98]. Based on the stress-and-coping model, when individuals appraise negative events such as resource scarcity as stressful, they consider their self-image as being threatened [99]. Prior studies have found that certain types of consumption may have an indirect positive effect. For instance, Townsend and Sood [100] find that choosing aesthetically appealing products influences self-worth, such as a decreased tendency to commit to failed actions and increased openness to counter-attitude arguments.

These studies suggest that choosing highly aesthetic products after experiencing self-discrepancy or threats resembles a traditional self-acceptance manipulation. Based on these findings above, we expect that self-acceptance, like compensatory consumption by HISC, may have a similar self-restoration function. Therefore, we expect that the propensity for HISC will be mitigated in individuals with self-acceptance when they are subjected to resource scarcity compared to individuals with low self-acceptance.

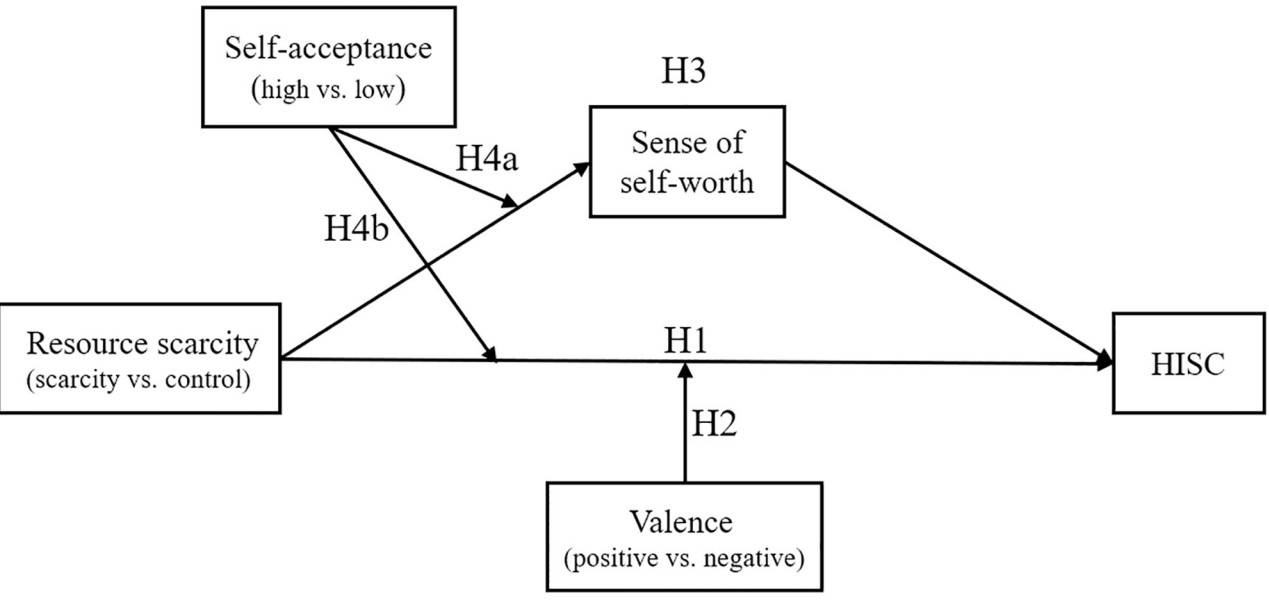

**Fig 1. Conceptual framework.**

**H4a:** Self-acceptance moderates the effect of resource scarcity on the sense of self-worth. Specifically, individuals with low self-acceptance who are subjected to resource scarcity have a lower sense of self-worth; while there was no significant difference between resource scarcity and sense of self-worth when individuals have high self-acceptance.

**H4b:** Self-acceptance moderates the impact of resource scarcity on HISC. Specifically, individuals with high self-acceptance who are subjected to resource scarcity will not exhibit a propensity for HISC compared to individuals with low self-acceptance.

Our conceptualization framework and hypotheses are summarized in Fig 1.

## 3. Study 1a

Study 1a was designed to test H1 by using linear regression method to study the effect of resource scarcity on consumers' audio consumption choices. Study 1a was conducted in a laboratory and recruited 96 participants from a business school in China. We measured participants' perceived resource scarcity in laboratory experiments. We examine the propensity for HISC by participants' attitude toward an intense (high decibel) sound and their preferred volume on continuous audio in the same audio clips ranging from less intense to high intense sound (low decibel to high decibel). The audio sound we used in study 1a is music clips without lyrics.

### 3.1 Methodology

**Participants and design.**   Ninety-six undergraduate students and teachers from the Business School participated in the experiment, including 49 females (51.04%) and 47 males (48.96%) ranging from 18 to 35-years-old with a mean age of 20 (SD = 2.421). Participants were told that they would be taking part in an unrelated study evaluating an audio clip being considered for a product advertisement and now needed to choose the appropriate volume for

this audio sound. First, we measured the perceived resource scarcity of participants on a four-item scale adapted from Roux and Goldsmith [9] (e.g., "My resources are scarce," "I don't have enough resources," "I need to protect the resources I have," and "I need to acquire more resources" (scale: 1 = Strongly disagree to 7 = Strongly agree)."). Responses to those four items were averaged to form an index of experienced resource scarcity (a = .95).

Participants were then asked to listen to a music clip without lyrics on computers using headphones in the computing laboratory. All participants used the same model and make of computer and headphones to listen to the music clips without lyrics to minimize differences in audio quality between individuals. Further, for the purposes of this study, brand new head-phones were purchased and tested by the assistant across all computers to ensure the same quality of audibility (the auditory equipment see Appendix A in S1 Appendix of S1 File).

The measurement of key dependent variables consisted of observing the volume levels selected by the participants while listening to the music clips. We take volume level as a measure of the dependent variable because physics studies define the sound intensity as the decibel level of a sound that best captures the intensity of a sound. In consumer behavior study, the decibel level is also used as a measure of sound intensity [7]. We used a decibel meter to record decibel levels at different volumes, and the record showed that the higher the volume, the higher the decibel level (the decibel meter see Appendix B in S2 Appendix of S1 File). Previous studies also show that the increased volume of auditory stimuli corresponds to increased decibel levels and more intensive sound [101]. The more saturated visual stimulus is more intensive, and sound of louder volumes is also more intensive.

At the start of the study, the volume level of all computers was set to "1". Participants were instructed to listen to a small part of the prepared music clip and then adjust the volume level of the headphones to a level that was comfortable for them. Computer settings allow participants to set the volume anywhere between 0 (mute) and 7 (maximum). They then listen to the audio clip again at the volume level of their choice. During this process the research assistant recorded the volume level selected by each participant. Finally, participants provided demographic information including gender, age, and education.

## 3.2 Results

**Dependent measures.**   Linear regression with audio clip volume as the dependent variable, resource scarcity as the independent variable, and age and gender as covariates revealed that none of the covariates was significant; therefore, they were not included in the following analyses. The linear regression revealed that a significant effect between resource scarcity and volume level (b = .752, t = 11.048, p = .00). The results show that resource scarcity positively affects HISC, that is, compared with resource-rich individuals, individuals subjected to resource scarcity chose higher-intensity volumes.

## 3.3 Discussion

Study 1a showed that subjects who were subjective to resource scarcity preferred to listen to high- volume music than subjects with abundant resources. Therefore, Hypotheses 1a is supported in the auditory domain. Study 1a provides an important finding, as most studies to date on compensatory consumption in response to resource scarcity have focused on the increased need for visual elements of consumption [37, 48]. Few studies have investigated the effects of resource scarcity on other sensory domains. However, Study 1a utilized only music clips, positively valenced auditory stimuli, and did not explore negatively valenced auditory stimuli. Study 1b will utilize the negatively valenced auditory. We expect that negatively valenced

auditory also lead to a preference for high-intensity volume in individuals subjected to resource scarcity (Hypothesis 2).

## 4. Study 1b

An important goal that study 1b was designed to test H2 by using PROCESS SPSS Mode l method to examine what effect the valence of HISC may have on the observed impacts. Study 1a was conducted in a laboratory and recruited 191 students and teachers from a business school in China. We manipulated participants' positively and negatively valenced experiences. Unlike study 1a, which used only positively valenced sensory stimuli, study 1b included negatively valenced stimuli. We expect that individuals subjected to resource scarcity prefer negatively valenced stimuli just as they prefer positively valenced stimuli, since negatively valenced stimuli also lead to higher levels of arousal, which in turn restores the self-discrepancy caused by resource scarcity (Hypothesis 2).

### 4.1 Methodology

**Pretest.**   Previous researchers have commonly used music to manipulate positive experiences [49, 102], and commonly used negative ambient sounds to operationalize negative experiences [7, 102, 103]. The pretest design consisted of 10 different audio clips and 40 participants (21 male) to identify a negatively and positively valenced clip for Study 1b. One minute of an instrumental clip of music was chosen for the positively valenced clip, which was used in study 1a, and one minute of the sound of a jet engine was chosen for the negatively valenced clip (1 = strongly disagree, 7 = strongly agree: "I like the clip," $M_{negative}$ = 2.33, SD = .662, vs. $M_{positive}$ = 5.71, SD = .929, t (78) = -18.626, p < .00).

**Participants and design.**   One hundred and ninety-one participants from college students and teachers were randomly assigned to two groups of positive and negative valence conditions. A total of 191 students and teachers participated in the experiment, including 93 females (48.70%) and 98 males (51.30%) ranging from 18 to 46-years-old with a mean age of 21.06 (SD = 4.34). 95 (48 females, 50.53%) participated in the negative valence conditions and 96 (47 females, i.e., 48.96%) in the positive valence conditions. As in experiment 1a, participants were told they would take part in a series of unrelated studies: one study completed survey questions about perceived resource scarcity and a second study selected audio clips that would be considered for the product advertisement.

As in Study 1a, after completing the resource scarcity survey questions, the participants took part in a background sound study described as a product advertisement and were asked to listen to the negatively or positively valenced audio clip in a low intensity condition of 5% maximum possible volume (low decibel volume) or a high intensity condition of 100% of maximum possible volume (high decibel volume). As in study 1a, all participants were equipped with the same laptops and headphones and were asked to listen to the audio clip, with the volume level and content of the audio clip being the only differences. After listening to a small part of the prepared audio clip, participants were allowed to adjust the volume level of headphones to a level between 0 (mute) and 7 (maximum) that was comfortable for them. The research assistant recorded the volume level selected by each participant. For experimental purposes, the research assistants checked that the volume levels selected by each participant had not been tampered with during the experiment. Finally, the participants provided demographic information including gender, age, and education level.

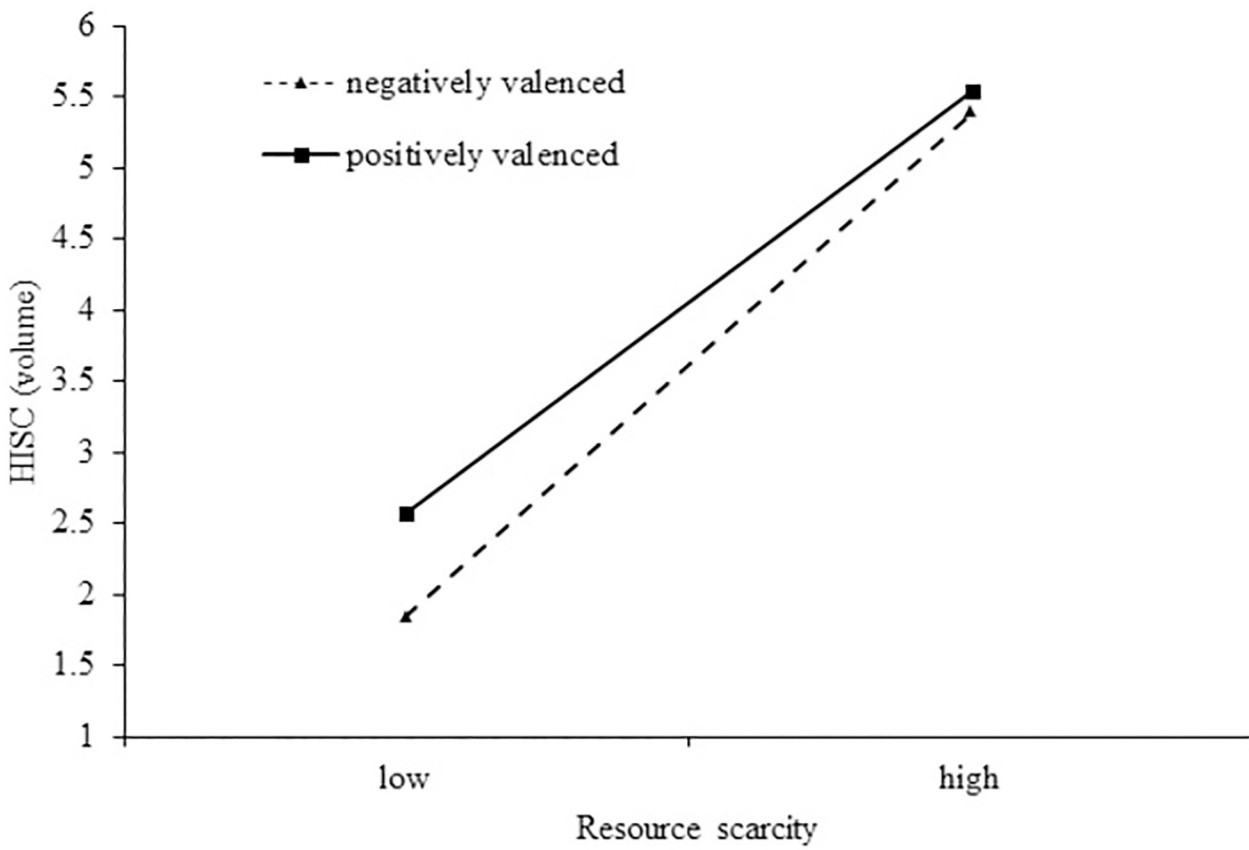

**Fig 2. Moderating effect of valence (negative vs. positive) on the resource scarcity and HISC.**

## 4.2 Result

**Dependent measures.** The main objective of Study 1b was to examine participants' preference for sensory stimulus intensity when participants were subjected to resource scarcity and exposed to different sound valences. The four items used to measure resource scarcity were combined into one composite measure (a = .94).

To test hypothesis 2, first, we performed the same analysis of results as in Study 1a on 95 participants (male = 46) who listened to negatively valenced audio clips. Linear regression with the volume of audio clip as the dependent variable and resource scarcity as the independent variable revealed a significant main effect between the resource scarcity and the volume level (b = .985, t = 15.404, p = .00). Compared to individuals whose resources are not scarce, individuals subjected to resource scarcity chose higher intensity volumes. Second, we tested the moderating role of different valences including positively and negatively valenced sensory stimuli on the main effects of resource scarcity on sensory intensity. We performed moderation analyzes, using the PROCESS SPSS macro [104] (Mode l 1; 5,000 bootstraps; Hayes, 2013) and did not obtain a significant interaction effect (b = -.944, SE = .0925, t = -1.02, p = 0.309, 95% CI = [-.2769, .0881]. The results also showed that resource scarcity had a significant main effect on HISC (b = .8660, SE = .0692, t = 12.5155, p = .00, 95% CI = [.7295, 1.0025]), but the audio type (positively valenced vs. negatively valenced) did not replicate this effect (b = .7046, SE = .4251, t = 1.6574, p = .0991, 95% CI = [-.1341, 1.5433]) (see Fig 2). These moderating

effects corroborate Hypothesis 2 and suggest that individuals subjected to resource scarcity simply prefer the intensity of sensory stimuli independent of valence.

### 4.3 Discussion

Study 1b showed that participants subjected to resources scarcity exhibit a propensity for a higher volume of negatively valenced audio clips than those without resources scarcity. HISC compensates for the self-discrepancy of individuals subjected to resource scarcity, and interestingly, we find that negatively valenced HISC can also restore this self-discrepancy. Thus, only intensity, not valence, influenced resource-scarce individuals' preferences for high-intensity sensory consumption. Therefore, hypotheses 1a and 1b are supported in different valence domains. Although the results of study 1a and study 1b document the role that HISC plays in resource-scarce individuals seeking compensatory consumption, one might wonder about the role mediating variables play.

## 5. Study 2

In studies 1a and 1b, we demonstrate that individuals subjected to resource scarcity exhibit a propensity for HISC in the auditory domains. We speculate that resource scarcity would reduce an individual's self-worth and that HISC itself, like other affirming activities, serves as a restorative mechanism of self-worth [105] (Steele 1988; Townsend, Sood 2012). The design of Study 2 is to test H3 by using PROCESS SPSS Mode l method to document the mediating role of self-worth between resource scarcity and HISC in the visual domain. Study 2 was conducted on an online research service platform (www.credamo.com) called Credamo and recruited 182 voluntary participants. We manipulated participants' resource scarcity in color sensory stimulant context. We hypothesize that resource scarcity reduces the individual's self-worth, which in turn promotes an individual's HISC preference and ultimately has a restorative effect on self-worth. Unlike studies 1a and 1b, in study 2 we examine the propensity for HISC by the attitude of the participants toward intense color (highly saturated) and by their preferred choice of color on a continuum of shades ranging from less intense (less saturated) to more intense colors (more saturated) in the same hue family.

### 5.1 Methodology

**Pretest.** In prior research, the intensity of a color was defined by its saturation level (or chromacity), which is a reflection of the amount of pigmentation that a color contains [106, 107]. High-chroma colors are intense, vivid, and strong, whereas low-chroma are weak and light. Research also find that adjusting the saturation and brightness of a color affects the level of arousal [26]. The pretest design consisted of two different intensities of orange color and 50 participants (23 male) to identify high-intensity and low-intensity sense stimuli for Study 2. Orange with a saturation level of 255 was selected as a high intensity sensory stimulant, and orange with a saturation level of 240 was selected as low intensity sensory stimuli (Fig 3), both with a hue of 28 (1 = does not describe this color at all, 7 = perfectly describes this color: bright, vibrant, bold and vivid, $M_{high\ intensity}$ = 5.97, SD = .66 vs. $M_{low\ intensity}$ = 2.72, SD = 1.18; t (48) = -12.03, p < .00). The color swatches of high- and low- intensity visual stimuli in our experiment were modulated by chemical engineers using professional instrument of colorimeters (the instrument see Appendix C in S3 Appendix of S1 File).

**Participants and design.** We recruited 182 voluntary participants from an online research service platform (www.credamo.com) called Credamo who were paid RMB 1.5 (USD 25 cents) for the online experiment. The final valid sample comprised 160 participants (82 female i.e., 51.3%, average age 31.34, ranging from 18 to 62 with SD = 9.648) who were randomly

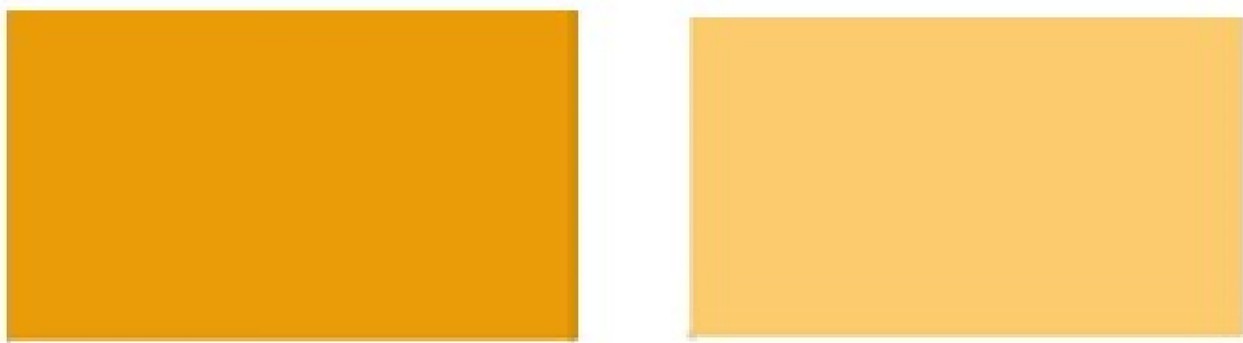

**Fig 3. Orange color swatches of high- and low- intensity visual stimuli.**

assigned to the conditions of resource scarcity and control condition, after excluding those incomplete responses. 80 (39 females, 48.75%) participated in resource scarcity conditions and 80 (43 females, that is, 53.75%) in control conditions. Among the 160 participants, 25 (15.63%) had master's degree, 99 (61.88%) had bachelor's degree and 36 (22.50%) had junior college degree or below. Participants first finished an episodic recall task, which is adapted from Fischhoff B. et al. [108]. Specifically, in the resource scarcity condition, participants were required to describe three to five episodes when they felt they "did not have enough of something" or "resources were scarce," such as having money or resources that could not meet their needs. They were next required to choose two of the episodes they just mentioned and state each of them in detail in no fewer than 50 words, explaining what was missing and what they went through. In the control condition, participants were first required to think about and write down three to five things they had done in the past week, then focus on and describe in detail two of these memorable events. To test whether episodic recall task activated perceptions of resource scarcity, participants were exposed to manipulation checks and then required to indicate the extent to which they agreed with the following items: "I don't have enough resources," "My resources are scarce," "I need to protect the resources I have," and "I need to acquire more resources" (scale: 1 = Strongly disagree to 7 = Strongly agree).

After completing the episodic recall task, the participants were asked to answer a series of questions aimed at measuring their self-worth. They reported their answers to the following statements (1 = completely disagree, 7 = completely agree): "I feel that I am on an equal plane with others," "I feel respect for myself," "I am a person of worth" and "I feel a sense of prestige," [109–111]. The participants also provided demographic information such as gender, age, and education.

Finally, participants were informed that the final study was a product decoration investigation that involved evaluating different shades of color for a product package. A virtual color sample of orange was presented to the participants without providing any other information, such as color name and brand of product. We only told the participants that the manufacturer was considering launching this color of packaging for their new products. The orange hue was chosen because prior research has shown that different hues vary in intensity and excitement properties. Hues such as blue have relaxing properties, while those such as orange have arousal and excitement properties [106, 107, 112, 113]. First, participants were asked to report their attitudes toward the color on a three-item scale (1 = completely disagree, 7 = completely agree): "This is a nice color," "I like this color," and "This is a good color." Then, a measure of

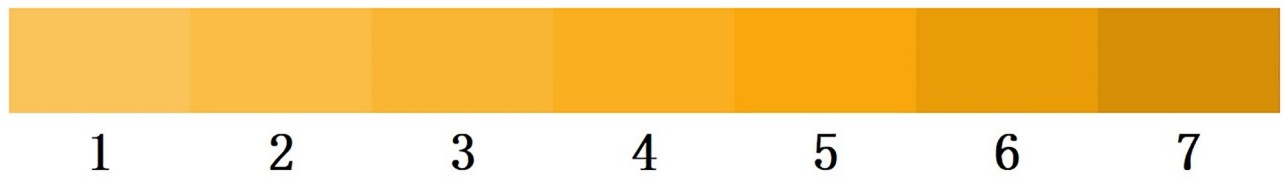

**Fig 4. Spectrum of low-high intensity color.**

dependent variable for HISC preference was collected. The experimental assistant showed the participants a set of seven shades of orange, with the last shade being the most intense shade (most saturated), and then the participants were asked to choose which shade they like the most. We manipulate this aspect of color based on prior research, such as 20% lower saturation in order for the six less intense hues on the scale (Fig 4). The intensity of a color is determined by its chroma, and colors of low chroma are weak and dull, while high-chroma colors are intense and strong [106, 107]. As in the pretest, participants were randomly asked to respond to an attribute numbered 1 or 7 representing different intensities of color. Participants were unaware that this measure was actually the study's true dependent measure.

During manipulation check of intensity of the stimulus, participants were invited to rate the color according to several attributes to measure its intensity and brightness (1 = did not describe the color at all, 7 = described the color perfectly): bright, vibrant, vivid, and bold. To eliminate individual factors as explanations for the findings, we also collected the inherent preference of participants for bright colors (1 = totally disagree, 7 = completely agree: "I prefer bright colors"). Finally, to rule out the effect of individuals with severe visual impairment, we required participants to state whether they had any of the following visual impairments: astigmatism/myopia/uncorrected hyperopia/cataract/ color blindness.

### 5.2 Results

**Manipulation checks.** Responses to the resource scarce item of manipulation checks were averaged to form the resource scarcity index ($\alpha$ = .91). Participants in the resource scarcity condition gave higher ratings on the experienced resource scarcity index ($M_{scarcity}$ = 6.03, SD = .54), compared to those in the control condition ($M_{control}$ = 2.98; SD = 1.19; F (1, 158) = 58.43, p $<$ .00), thus proving that the manipulation was effective. Responses to manipulation check items for color intensity were averaged to form an attribute index ($\alpha$ = .93), with higher attribute scores for saturated colors, relative to light colors. Orange in saturated color condition received higher scores on the intensity attributes index ($M_{high\ intensity}$ = 5.64; SD = .60; F (1, 158) = .27, p $<$ .00), compared to those in the light color condition ($M_{low\ intensity}$ = 2.35, SD = .79), thus providing evidence that the manipulation of color intensity was effective.

**Dependent measures.** To further test the theory proposed by our (hypothesis 3), we tested the meditational role of self-worth in explaining the relationship between resource scarcity and HISC. Using the PROCESS SPSS macro [104] (Model 4; 5,000 bootstraps; Hayes, 2013) method, we conducted mediation analyzes. We entered resource scarcity as the independent variable, self-worth as the mediator variable, and sensory intensity as the dependent variable. Gender, age, and preference for orange were entered as covariates, and none of the covariates were significant and thus was dropped from the analyses. A closer look at the bootstrap results supporting our theorized process (H3), the mediation of self-worth was found to be significant because of the indirect effect (b = -1.32, 95% CI [−1.90, -.76]). Furthermore, the direct effect was also significant (b = -1.51, 95% CI: [-2.10, -.91]), suggesting that the impact of resource

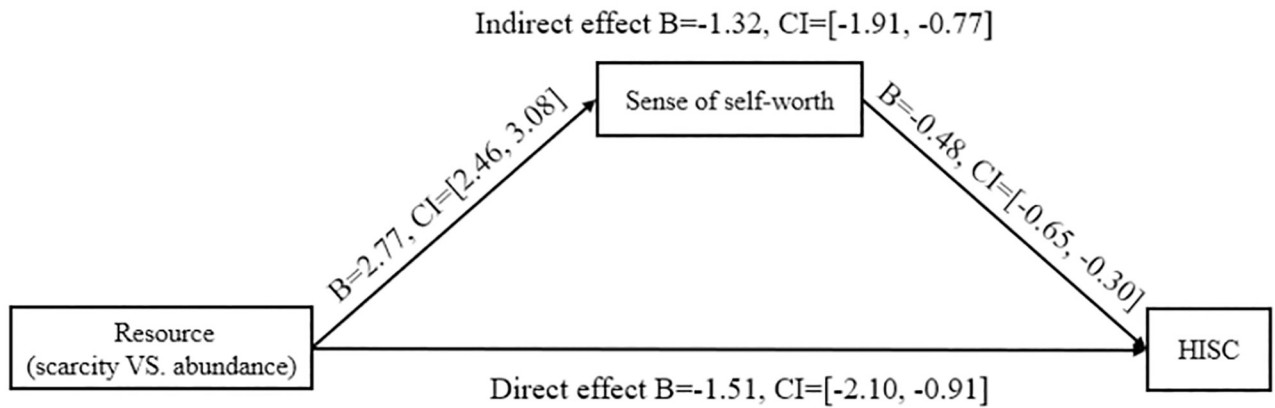

**Fig 5. The mediating effect of sense of the self-worth.**

scarcity on HISC was partly mediated by the sense of self-worth (Fig 5). These results support our hypothesis 3 and suggest that individuals subjected to resource scarcity increase HISC preferences through the mediation of self-worth for self-restoration.

### 5.3 Discussion

Study 2 shows that individuals who were subjected to resource scarcity have lower self-worth and therefore prefer HISC relative to those who do not suffer from resource scarcity. Study 2 not only supports Hypothesis 1 again in the visual domain, but also shows that the preference for HISC in individuals subjected to resource scarcity is mediated by self-worth, thus supporting Hypothesis 3. Hypotheses 1 and 2 were supported in a different sensory domain. Studies 1 and 2 show that compensatory consumption in response to resource scarcity by individuals occurs WITH a heightened demand for auditory and visual elements of consumption. Very few studies have examined the impact of resource scarcity on demand in the tactile sensory domain. Since the preference for HISC by individuals subjected to resource scarcity is mediated by self-worth, one may wonder whether participants subjected to resource scarcity and provided with additional self-worth would still prefer HISC. Study 3 aimed to expand the findings of Study 2 by checking whether this propensity for HISC is also evident in individuals with self-acceptance as well as in the tactile sensory domains.

## 6. Study 3

The design purpose of Study 3 is to test H4 by using PROCESS SPSS Mode 8 to investigate the moderating effect of self-acceptance on resource-scarce individuals' preference for HISC in the tactile domain. Study 3 was conducted on an online research service platform (www.credamo.com) called Credamo and recruited 251 voluntary participants. We manipulated participants' resource scarcity and self-acceptance in a tactile sensory experience. Study 3 shows that individuals subjected to resource scarcity prefer HISC in the visual domain when they are not allowed additional opportunities to restore their self-worth, which is mediated by self-worth. First, Study 3 is designed to investigate the robustness of the effects observed in previous studies by testing whether resource scarcity impacts the preference for HISC in the tactile domain. Second, Study 3 will expand on previous studies by testing whether self-acceptance attenuates the mediating effect of Study 3.

## 6.1 Methodology

**Participants and design.**   Experiment 3 was still conducted on the Credamo platform (www.credamo.com), with a total of 251 voluntary participants who were paid RMB 1.5 (USD 25 cents) for the online experiment. The final valid sample comprised 240 participants (154 female i.e. 64.17%, average age 29.21, ranging from 18 to 59 with SD = 7.836) who were randomly assigned to the conditions of a 2 (resource: scarcity vs. abundance) x 2 (self-acceptance: high vs. low) between-subjects design. Among the 240 participants, 38 (15.80%) had master's degree, 154 (64.20%) had bachelor's degree and 48 (20%) had junior college degree or below. The measurement for self-worth were identical to those used in study 2. Participants were still told that they would participate in several unrelated studies. According to the manipulation procedure in Briers and Laporte [18], the research assistant first asked participants to state the total amount of their savings on the nine-point scale, with a response scale manipulated: participants in the resource scarcity condition offered their rating by using a scale anchored from 1 ("0–5,000 RMB") to 9 ("over 4,000,000 RMB"), while participants in the resource abundance condition offered their rating by using a scale anchored from 1 ("0–5,0 RMB") to 9 ("over 400 RMB"). The participants in the resource scarcity (vs. abundance) condition were then asked to write a short essay of no less than 50 words about what it would be like to live in a poor (vs. rich) family. Twenty-eight participants did not complete the essay task following the requirements and were excluded from subsequent analyzes. We used the same manipulation test as in Study 2 to test whether the resource-scarce state of the participants was activated.

After completing the savings rating study, participants were asked to participate in an exercise designed to manipulate participants" self-acceptance. Following the manipulation exercise in Kim and Gal [93] (2014), participants under the high self-acceptance condition were required to read over a list of "thoughts" that can help increase self-acceptance (e.g., "I would better not define myself entirely by my behavior, by others" opinions, or by anything else under the sun."; "I can accept myself whether I win, lose, or draw," see Appendix D in S4 Appendix of S1 File for a complete list of those thoughts) and were given a chance to choose their favorite thought. They were then asked to explain in no less than 50 words in detail why this particular chosen thought was their favorite, and how this thought could be applied to their daily lives. Those in the low self-acceptance condition were asked to read over a list of daily activities (e.g., "listen to music," "Eating breakfast"; see Appendix D in S4 Appendix of S1 File for a complete list of activities), to choose one of their favorite daily activities, and to explain in no less than 50 words why that particular activity was their favorite. As a manipulation check for self-acceptance levels in the two conditions, participants were asked to answer to what extent they agreed with the statements on a seven-item scale (1 = strongly disagree, 7 = totally agree): e.g., "It's unbearable to fail at important things, and I can' t stand not succeeding at them", see Appendix E in S5 Appendix of S1 File [114, 115].

After completing the same self-worth manipulation as in study 2, participants were asked to take part in a HISC preference experiment in the tactile domain. Krishna & Morrin [116] find that consumers' tactile feelings on products or their packaging will directly affect consumers' evaluation of products. Consumers have great differences in the intensity of the tactile experience. Those consumers who are more inclined to get product information by touching products pay more attention to tactile enjoyment and psychological compensation of products than ordinary consumers [117–120]. According to the PECK [118] study, participants were asked to complete an experiment designed to measure differences in the intensity of individuals' preference for touch on a 12-item scale (1 = completely disagree, 7 = completely agree): "When walking through stores, I can't help touching all kinds of products," "I find myself touching all kinds of products in stores," and "Touching products can be fun."; see Appendix

**Table 1. Moderation test.**

| | Sense of self-worth | | | HISC | | |
|---|---|---|---|---|---|---|
| | **Coeff** | **Se** | **t** | **Coeff** | **Se** | **t** |
| constant | 2.29 | 0.33 | 6.99*** | 5.89 | 0.3 | 17.21*** |
| Resource scarcity | 2.89 | 0.13 | 22.14*** | -1.60 | 0.19 | -8.28*** |
| self-acceptance | 3.28 | 0.13 | 25.32*** | -2.03 | 0.21 | -9.62*** |
| sense of self-worth | | | | -0.25 | 0.06 | -4.49*** |
| Int_1 | -2.91 | 0.18 | -15.84*** | 2.1 | 0.22 | 9.44*** |
| R-sq | 0.80 | | | 0.79 | | |
| F | 150.90 | | | 122.15 | | |

Note: ***, **, * represent significant levels at 1%, 2% and 5%, respectively.

Int_1: Resource scarcity × self-acceptance

F in S6 Appendix of S1 File for the full set of 12-items). Finally, demographic information such as the age of the subjects was collected.

## 6.2 Results

**Manipulation checks.** Responses to the resource scarce item of manipulation checks were averaged to form the resource scarcity index (a = .89). Participants in the resource scarcity condition gave higher ratings on the experienced resource scarcity index ($M_{scarcity}$ = 5.89, SD = .62), compared to those in the resource abundance condition ($M_{abundance}$ = 2.26; SD = .81; $F$ (1, 238) = .69, p < .00), thus proving that the manipulation was effective. The 7-item scale used to measure self-acceptance had high reliability (α = 0.89). Participants in the low self-acceptance condition provided higher ratings on the self-acceptance index ($M_{high}$ = 5.23, SD = .60), compared to those in the high self-acceptance condition ($M_{high}$ = 2.95; SD = .86; $F$ (1, 238) = 11.47, p < .00), thus the manipulation of self-acceptance worked.

**Moderated mediation.** To test our proposed hypothesis 3, we examined the moderating role of self-acceptance in explaining the relationship between resource scarcity and HISC. Using the PROCESS SPSS macro (Model 8; 5,000 Bootstrap; Hayes, 2013) approach, we performed moderator variable analyzes. We used resource scarcity as an independent variable, self-acceptance as a moderator variable, self-worth as a mediator variable, and perceived intensity as a dependent variable. The results, as shown in Table 1, revealed that the interaction between resource scarcity and self-acceptance had a significant impact on sense of self-worth (B = -2.91, $t_{(238)}$ = -15.81, p < .01, 95% CI: [-3.2754, -2.5496]) and HISC (B = 2.13, $t_{(238)}$ =— 9.70, p < .01, 95% CI: [1.70, 2.57]). This indicates that self-acceptance can not only moderate the effect of resource scarcity on HISC, but also moderate the effect of resource scarcity on the sense of self-worth. Specifically, moderated mediation revealed a significant mediation effect of self-worth for low self-acceptance participants (b =—.68, SE = .16, 95% CI = [-1.02, -.38]), but not for high self-acceptance participants (b = .01, SE = .03, 95% CI = [-.04, .06]). This indicates that the self-worth only mediates the effect for the low self-acceptance participants.

Further simple slope analysis on self-worth shows that, as depicted in Fig 6, when the level of self-acceptance is low, the resources of individuals (scarcity vs. abundance) have a significant positive impact on self-worth (simple slope = 2.89, se = .13, t = 22.14, p <0. 01, CI: [2.64, 3.15]; while when self-acceptance is high, individuals' resources (scarcity vs. abundance) have no significant effect on sense of self-worth (simple slope = -.016, se = .13, t = -.12, p = 0.90, CI: [-.27, .24]). It can be seen from Fig 6 that under the condition of low self-acceptance, the more

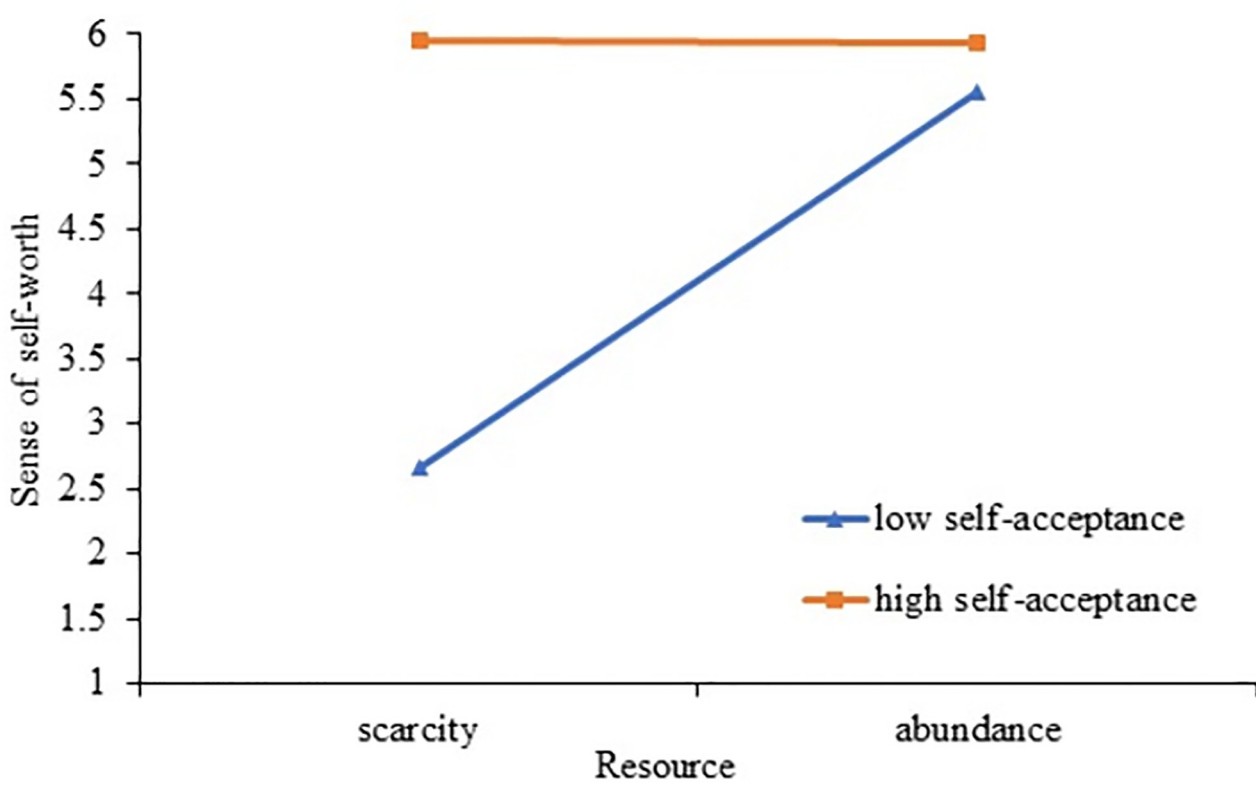

**Fig 6. Moderating effect of self-acceptance on the resource scarcity and self-worth.**

scarce resources an individual has, the lower his sense of self-worth is, while this effect disappeared under conditions of high self-acceptance. The results show that the level of self-acceptance moderates the effect of resource scarcity on sense of self-worth, and H4a is supported. It can be seen from Fig 7, when the level of self-acceptance is low, the resources of individuals (scarcity vs. abundance) have a significant negative impact on HISC (simple slope = -1.60, se = .19, t = -8.28, p<0. 01, CI: [-1.98, -1.22]; while when self-acceptance is high, individuals' resources (scarcity vs. abundance) have a significant positive impact on HISC (simple slope = .50, se = .11, t = 4.56, p<0.01, CI: [.28, .71]). Fig 7 also shows that individuals subjected to resource scarcity do not show a preference for HISC in the case of high self-acceptance compared to the case of low self-acceptance. The results suggest that the level of self-acceptance moderates the effect of resource scarcity on HISC, and H4b is supported.

## 6.3 Discussion

Participants subjected to resource scarcity and who were low self-acceptance rated intensity of touch needs significantly higher than those also subjected to resource scarcity and who were high self-acceptance. On the contrary, the evaluations of the intensity of the touch needs did not differ significantly among the participants who were abundance in resource, regardless of whether self-acceptance was high or low. When allowed to freely choose a preferred intensity of touch needs from a continuum of less to more intense needs, we again find that only the participants subjected to resource scarcity, low self-acceptance condition selected a

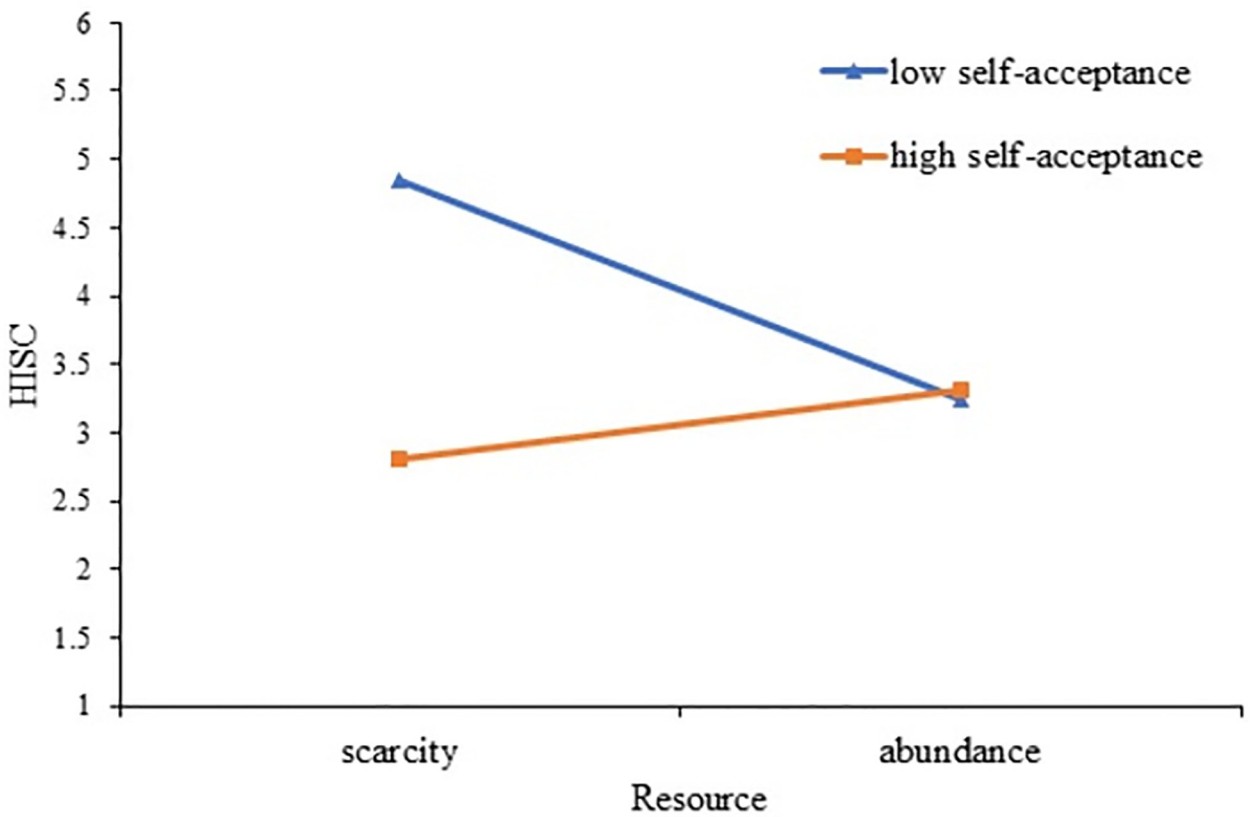

**Fig 7. Moderating effect of self-acceptance on the resource scarcity and HISC.**

significantly more intense touch needs, without significant difference in the choices of the other three conditions. Therefore, Hypothesis 3 is supported.

Study 3 showed a propensity of individuals who were subjected to resource scarcity to consume more intense touch sensory stimuli in situations of low self-acceptance. However, those individuals with high self-acceptance and subjected to resource scarcity did not exhibit the same propensity for HISC. Therefore, Study 3 not only validated the moderating effect of self-acceptance, but also extended the findings of Study 1 and Study 2 from the haptic domain.

## 7. General discussion

### 7.1 Theoretical contribution

Across four studies, we find that individuals who are subjected to resource scarcity show a preference for high-intensity sensory consumption in the auditory, visual, and tactile domains, as demonstrated through a higher propensity for louder volume levels of music, more intense (saturated) colors, and more intense need for touch. We also find that both negatively and positively valenced sensory stimuli have the same impact on resource-scarce individuals' preference for HISC. That is to say, only intensity, not valence, impacts resource-scarce individuals' preferences for high-intensity sensory consumption. Furthermore, we demonstrate that the sense of self-worth significantly mediates the effect of resource scarcity on HISC. Finally, we

reveal a boundary condition for the effect and find that the preferences for HISC is lower following resource-scarce individuals who engage in a self-acceptance.

This study contributes to the extant literature in the following three important ways. First, we contribute to a literature on sensory consumption by demonstrating the novel application of three basic sensory consumptions, without symbolic cues such as social status or brand, in resource-scarce conditions. Second, we add to the literature on resource scarcity and compensatory consumption in several ways. We suggest not only that HISC can be used as a form of reactive consumption after resource scarcity, but also identify the mediating mechanism of self-worth. Previous research has discussed the concept of reactive consumption as a means for escaping from self-awareness and as being distractive [20, 121, 122], while being agnostic about the process by which self-restoration due to resource scarcity may occur. Although there is literature that examines the effect of resource scarcity on counter hedonic consumption or consumer preference for options [31, 48], there are few studies on the effect of resource scarcity on HISC. We identify a causal pathway for the effect of a new form of reactive consumption on resource scarcity by showing that resource scarcity reduces individuals' sense of self-worth and thus prefers HISC. In this sense, we provide an understanding of when and how distractive consumption may lead to self-restoration under resource-scarce conditions. Third, although the role of HISC in mood management and healthy strategies has been examined in prior literature [22, 123, 124], it has not been specifically studied in the context of resource scarcity management. An important understanding of HISC as compensatory consumption may be extended to other forms of reactive compensatory consumption studied in the literature. We demonstrate that at least some types of HISC (eg, auditory, visual, and tactile) can have a compensatory effect on the reduction in self-worth following resource scarcity. We also find the self-acceptance boundary condition for this effect.

## 7.2 Substantive implications

Our research will have some important implications at the consumer level. Given that individuals subjected to resource scarcity have choices in different sensory domains for HISC preferences, consumers can engage in some HISC that are beneficial to them and avoid some HISC that are harmful to them. Although prior literature suggests that indulging in high-intensity sensor consumption in the gustatory domain can be distracting, it may also have detrimental effects on one's health. Our findings suggest that the consumer can achieve a similar compensatory effect by engaging in the HISC stimuli in any other sensory domain. Such HISC may include indulging in shopping products with saturated (intensive) colors, listening to loud music, or enjoying a deep-tissue massage. These forms of HISC can help restore a person's self-discrepancy due to resource scarcity without the detrimental effects of eating high-sugar foods.

Marketers and retailers might also benefit from our findings. It is well known that self-acceptance is an effective way for self-restoration, which is borne out by both the present study and prior research. Marketers can tailor and target their marketing policy of HISC toward consumer segments that may be more receptive or unreceptive to information based on the level of consumer self-acceptance. For example, certain populations (e.g., women, elderly) might be more self-accepting than others, and thus less likely to prefer HISC when subjected to resource scarcity. However, some populations with low self-acceptance are more open to HISC and marketers can pitch related products.

Given that extant research has demonstrated that consumers have higher willingness to pay for products after being touched [125]. Our research suggests that many sensory outcomes previously thought to be intrinsic dispositions [126], such as the need for touch, might also be

impacted by contextual influences. This contextual influence may be under the control of marketers, providing an avenue for retailers or marketers to influence consumers, for example, by subtly manipulating the level of "need for touch" in the stores. If consumers suffer from resource scarcity, they may be more proactive in touching or purchasing products due to the presence of products with an increased need for touch stimulus in store displays and advertisements.

In addition, consumers' preferences for HISC are high in resource-scarce circumstances such as the financial impacted by the COVID-19 pandemic. Our research also suggests that firms can increase sensory intensity in product design to meet consumer preferences, for example, by letting consumers choose the color intensity of a product when ordering online. This can also alleviate the negative impact of the epidemic to some extent.

### 7.3 Limitations and future research

Although we tested our hypothesis only in the auditory, visual and touch domains, we speculate HISC to have the same self-restorative effects in the taste and smell domains. Future research could examine whether individuals subjected to resource scarcity will have a higher preference for stronger perfumes or spicier foods. Di Muro and Murray [127] find that higher-tempo music and stronger scented perfumes result in higher arousal than slower-tempo music or milder-scented perfumes. In the future, researchers can also conduct different manipulations of the dependent variable measures for the sensory effects documented by ours. For example, although a propensity for louder music demonstrates that consumers who are subject to resource scarcity prefer high-intensity auditory consumption, another measure can be to observe whether such individuals also prefer music that is "richer or higher-tempo," as compared to those who do not suffer from resource scarcity. Finally, future studies of HISC preferences could investigate this phenomenon under more naturalistic conditions. Compared with the eastern part of China, the western part of China is economically underdeveloped and resources are relatively scarce. Minority groups in the West prefer to use brighter colors and shinier materials in traditional clothing and accessories. A major source of self-worth is an individual's socioeconomic status. Could it be that the long-term reduction in sense of self-worth due to low status impacts the level of sensory intensity sought by these consumers, driving a preference for visually more conspicuous products? This future study may contribute to a better understanding of the consumption dynamics of ethnic minorities and ethnic classes that are disadvantaged relative to the general population.

## Supporting information

**S1 File.**
(ZIP)

## Author Contributions

**Conceptualization:** Liangjun Peng, Yuxin Peng.

**Data curation:** Liangjun Peng, Haiyan Luo.

**Formal analysis:** Liangjun Peng, Yuxin Peng, Haiyan Luo.

**Funding acquisition:** Liangjun Peng.

**Investigation:** Liangjun Peng, Yeying Deng.

**Methodology:** Liangjun Peng, Yuxin Peng.

**Resources:** Haiyan Luo, Yeying Deng.

**Visualization:** Liangjun Peng.

**Writing – original draft:** Liangjun Peng, Yeying Deng.

**Writing – review & editing:** Liangjun Peng, Yuxin Peng.

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
