## [Decision Letter · Decision Letter 0]

15 Mar 2023

PONE-D-22-28511How High-Intensity Sensory Consumption Fill up Resource Scarcity: The Boundary Condition of Self-acceptancePLOS ONE

Dear Dr. Liangjun Peng,

Thank you for submitting your manuscript to PLOS ONE. After careful consideration, we feel that it has merit but does not fully meet PLOS ONE’s publication criteria as it currently stands. Therefore, we invite you to submit a revised version of the manuscript that addresses the points raised during the review process.

We look forward to receiving your revised manuscript.

Kind regards,

Abdullah Al Mamun, PhD

Academic Editor

PLOS ONE

Journal Requirements:

2. During your revisions, please note that a simple title correction is required: The word "fill" should be replaced with "fills". Please ensure this is updated in the manuscript file and the online submission information.

3. Please provide additional details regarding participant consent. In the Methods section, please ensure that you have specified (1) whether consent was informed and (2) what type you obtained (for instance, written or verbal). If your study included minors, state whether you obtained consent from parents or guardians. If the need for consent was waived by the ethics committee, please include this information.

Additional Editor Comments (if provided):

Please check the reviewer(s) comments carefully, then amend (please do not submit half-baked revision) and resubmit the manuscript accordingly.

Reviewers' comments:

Reviewer's Responses to Questions

**Comments to the Author**

1. Is the manuscript technically sound, and do the data support the conclusions?

Reviewer #1: Yes

Reviewer #2: Yes

2. Has the statistical analysis been performed appropriately and rigorously? 

Reviewer #1: Yes

Reviewer #2: Yes

3. Have the authors made all data underlying the findings in their manuscript fully available?

Reviewer #1: Yes

Reviewer #2: Yes

4. Is the manuscript presented in an intelligible fashion and written in standard English?

Reviewer #1: Yes

Reviewer #2: Yes

5. Review Comments to the Author

Reviewer #1: 1) The DOE was quite interesting having said that the auditory equipment and color intensity instrument were lacking.

2) This study would be even better with a proper instrumentation as this could be collaborated with other transdisciplinary personnel (pity it did not happen). I would possibly reject this paper as it is lacking of that simple arrangement. However, other angle is considered quite interesting to taken into consideration as this might be explore even more.

3) The sampling distribution are quite unacceptable and the participant involved could be classified into generic parameters (e.g. educational background/traumatic event/sex/races/ages/lifestyle etc.).

4) What is your control sampling?

Reviewer #2: The paper entitled: “How High-Intensity Sensory Consumption Fill up Resource Scarcity: The Boundary Condition of Self-acceptance” reports research showing the effect of resource scarcity on peoples HISC. The authors include data of four studies - presenting self-worth as mediator and self-acceptance as moderator. Also, they show the effect of scarcity on HISC on several sensory dimensions. The paper is well-written and informative. There is a notable progress across the studies and the validity of the effect is supported through the usage of the different methods. The analyses are adequate, and the data supports the hypotheses. However, there are also some issues that the authors need to address when revising their paper.

The moderated mediation in Study 3 needs more explanation and the results should be described in more details. Currently, there are only explanations regarding the moderation effects. What is important to know here is if the self-worth only mediates the effect for the low-acceptance subsample?

Generally, it is possible that there is a risk for a certain tautology when using self-worth and self-acceptance as mediator and moderator. The authors should explain the difference in more detail and discuss how it is possible that there is a strong connection but still room for these different processes.

The authors should reflect and discuss about the relation/processes regarding the self-worth restauration and the HISC with negatively valanced stimuli. For the stimulation with positive sensory input the effect is easily understandable, but it is quite difficult to see a reason why people experiencing scarcity leading to fewer self-worth should be interested in increasing their arousal through negative valanced stimuli.

Minor points

The methods are changed throughout the course of the research. Sometimes just one clip is shown, and participants chose their volume level – sometimes they start with high vs. low intensity volume – why did the authors change the procedure? Please include some explanations.

Pease include standardized betas in the results sections.

There are several differences/errors in the way the authors cite the references. Please correct this.

Some minor typos need to be corrected as well.

6. PLOS authors have the option to publish the peer review history of their article (what does this mean?). If published, this will include your full peer review and any attached files.

Reviewer #1: No

Reviewer #2: **Yes: **Anne Berthold

---

## [Author Response · Author response to Decision Letter 0]

30 Mar 2023

Response to Reviewer 1

Reviewer#1, Concern # 1: The design of experiment (DOE) was quite interesting having said that the auditory equipment and color intensity instrument were lacking.

Author Response: In Section 3.1 Methodology, we explain that all participants used the same model and make of computer and headphones to listen to the music clips without lyrics. So, the auditory equipment used in our experiments were computers and headphones. In the revised manuscript, we added auditory equipment. The color swatches of high- and low- intensity visual stimuli in the experiment were indeed modulated by chemical engineers using professional colorimeters. We did not emphasize this process in the first manuscript. Thank the reviewer for finding that we are missing the description of colorimeter. In the revised draft, we added the description of colorimeter which was omitted in the first manuscript.

Author action: 

(1) In Section 3.1 Methodology, we update the manuscript as follows: Further, for the purposes of this study, brand new headphones were purchased and tested by the assistant across all computers to ensure the same quality of audibility (the auditory equipment see Appendix A).

(2) In Section 5.1 Methodology, we update the manuscript as follows: The color swatches of high- and low- intensity visual stimuli in our experiment were modulated by chemical engineers using professional instrument of colorimeters (the instrument see Appendix B) 

Reviewer#1, Concern # 2: This study would be even better with a proper instrumentation as this could be collaborated with other transdisciplinary personnel (pity it did not happen). I would possibly reject this paper as it is lacking of that simple arrangement. However, other angle is considered quite interesting to taken into consideration as this might be explored even more.

Author Response: As mentioned earlier, we used some necessary equipment and instruments in our experiment, such as colorimeter, decibel meter, computer, earphone, etc., which are not emphasized in the manuscript. We collaborate with other interdisciplinary people to use these devices, like color palette engineers and audiologists. In the revised manuscript, we have added the missing equipment, such as colorimeter, decibel meter, computer, and earphone, and illustrated the equipment drawings in Appendix A, B, and C.

Author action: In Section 3.1 Methodology, we update the manuscript as follows: We used a decibel meter to record decibel levels at different volumes, and the record showed that the higher the volume, the higher the decibel level (the decibel meter see Appendix C). Previous studies also show that the increased volume of auditory stimuli corresponds to increased decibel levels and more intensive sound (Gelfand 2009).

Reviewer#1, Concern # 3: The sampling distribution are quite unacceptable and the participant involved could be classified into generic parameters (e.g., educational background/traumatic event/sex/races/ages/lifestyle etc.).

Author Response: We consider the validity of the sample distribution in the experiment and we collect demographic data about the participants in each experiment. In the revised manuscript, the the participants involved have been classified into generic parameters (eg, educational background/sex/ages/major., etc.).

Author action: 

(1) At the section 3.1 Methodology, we updated the manuscript as follows: Ninety-six undergraduate students and teachers from the Business School participated in the experiment, including 49 females (51.04%) and 47 males (48.96%) ranging from 18 to 35-years-old with a mean age of 20 (SD=2.421).

(2) At the section 4.1 Methodology, we updated the manuscript as follows: One hundred and ninety-one participants from college students and teachers were randomly assigned to two groups of positive and negative valence conditions. A total of 191 students and teachers participated in the experiment, including 93 females (48.70%) and 98 males (51.30%) ranging from 18 to 46-years-old with a mean age of 21.06 (SD=4.34). 95 (48 females, 50.53%) participated in the negative valence conditions and 96 (47 females, i.e., 48.96%) in the positive valence conditions.

(3) In Section 5.1 Methodology, we updated the manuscript as follows: We recruited 182 voluntary participants from an online research service platform(www.credamo.com) called Credamo who were paid RMB 1.5 (USD 25 cents) for the online experiment. The final valid sample comprised 160 participants (82 female i.e., 51.3%, average age 31.34, ranging from 18 to 62 with SD=9.648) who were randomly assigned to the conditions of resource scarcity and control condition, after excluding those incomplete responses. 80 (39 females, 48.75%) participated in resource scarcity conditions and 80 (43 females, that is, 53.75%) in control conditions. Among the 160 participants, 25 (15.63%) had master's degree, 99 (61.88%) had bachelor's degree and 36 (22.50%) had junior college degree or below. 

(4) In Section 6.1 Methodology, we update the manuscript as follows: Experiment 3 was still conducted on the Credamo platform (www.credamo. com), with a total of 251 voluntary participants who were paid RMB 1.5 (USD 25 cents) for the online experiment. The final valid sample comprised 240 participants (154 female i.e. 64.17%, average age 29.21, ranging from 18 to 59 with SD=7.836) who were randomly assigned to the conditions of a 2 (resource: scarcity vs. abundance) x 2 (self-acceptance: high vs. low) between-subjects design. Among the 240 participants, 38 (15.80%) had master's degree, 154 (64.20%) had bachelor's degree and 48 (20%) had junior college degree or below.

Reviewer#1, Concern # 4: What is your control sampling?

Author Response: Thank you for your comment on the control sampling in our manuscript. We appreciate the opportunity to clarify this aspect of our study.

(1) We selected the control group from a population of individuals who were matched to the study group based on educational background, age, sex, and other demographic factors.

(2) The control group consisted of individuals who were similar to the experimental group in terms of educational background, age, gender, and other relevant factors. This allows us to determine whether any changes observed in the experimental group are due to antecedent variables or other factors.

(3) Eligibility for the control group was determined based on the absence of any other significant factor conditions that could potentially confound the results.

(4) We recognize the importance of using an appropriate control group to ensure the validity and reliability of our findings, and we are confident that the control sampling method we used is appropriate for the research question being investigated.

(5) We will provide the following additional details and justification for this method in the revised manuscript to further clarify the study design.

Author action: 

(1) In the section 3.1 Methodology, we updated the manuscript as follows: Ninety-six undergraduate students and teachers from the Business School participated in the experiment, including 49 females (51.04%) and 47 males (48.96%) ranging from 18 to 35-years-old with a mean age of 20 (SD=2.421).

(2) In the section 4.1 Methodology, we updated the manuscript as follows: One hundred and ninety-one participants from college students and teachers were randomly assigned to two groups of positive and negative valence conditions. A total of 191 students and teachers participated in the experiment, including 93 females (48.70%) and 98 males (51.30%) ranging from 18 to 46-years-old with a mean age of 21.06 (SD=4.34). 95 (48 females, 50.53%) participated in the negative valence conditions and 96 (47 females, i.e., 48.96%) in the positive valence conditions.

(3) In Section 5.1 Methodology, we updated the manuscript as follows: We recruited 182 voluntary participants from an online research service platform(www.credamo.com) called Credamo who were paid RMB 1.5 (USD 25 cents) for the online experiment. The final valid sample comprised 160 participants (82 female i.e., 51.3%, average age 31.34, ranging from 18 to 62 with SD=9.648) who were randomly assigned to the conditions of resource scarcity and control condition, after excluding those incomplete responses. 80 (39 females, 48.75%) participated in resource scarcity conditions and 80 (43 females, that is, 53.75%) in control conditions. Among the 160 participants, 25 (15.63%) had master's degree, 99 (61.88%) had bachelor's degree, and 36 (22.50%) had junior college degree or below. 

(4) In Section 6.1 Methodology, we update the manuscript as follows: Experiment 3 was still conducted on the Credamo platform (www.credamo. com), with a total of 251 voluntary participants who were paid RMB 1.5 (USD 25 cents) for the online experiment. The final valid sample comprised 240 participants (154 female i.e., 64.17%, average age 29.21, ranging from 18 to 59 with SD=7.836) who were randomly assigned to the conditions of a 2 (resource: scarcity vs. abundance) x 2 (self-acceptance: high vs. low) between-subjects design. Among the 240 participants, 38 (15.80%) had master's degree, 154 (64.20%) had bachelor's degree and 48 (20%) had junior college degree or below.

Response to Editor

Author Response: We referenced the PLOS ONE style template to ensure that the manuscript met PLOS ONE style requirements. For example, we revised the file naming and abstract as follows.

How high-intensity sensory consumption fills up resource scarcity: The boundary condition of self-acceptance

Abstract:

Objective

Everyone in life will experience resource scarcity, which causes self-discrepancy. It is widely known that individuals participate in reactive consumption to solve the problems of self-discrepancy and resources scarcity. This kind of consumption may be symbolically related to the essence of the resource scarcity or may occur in an unrelated domain. This study proposes a theory for "filling up" one's resource scarcity through high-intensity sensory consumption (HISC).

Methods

According to the theoretical development, we put forward five hypotheses, which were verified by SPSS one-way ANOVA, mediation effect and moderating effect. Four experiments in the study were conducted from May 2022 and August 2022 and involved undergraduates from a university and volunteers recruited online. All participants are adults and verbally agree to participate voluntarily. Participants were grouped according to resources (scarcity vs. abundance) and self-acceptance (high vs. low). Data were collected with the scarcity questionnaire, the self-worth scale for adults, the self-acceptance questionnaire, and the high-intensity sensory preference scale responded by participants.

Results

Four studies suggest that not only do individuals facing resources scarcity prefer HISC but also that this consumption is mediated and moderated by self-worth and self-acceptance, respectively. This preference for HISC is negated when individuals have high self-acceptance traits. The findings are tested in the auditory domain (as evidenced by a propensity for louder volume), the visual domain (as evidenced by a propensity for more intense colors), and the tactile domain (as evidenced by a propensity for more intense need for touch). The findings also demonstrate that individual preferences for HISC is shown to operate regardless of the valence (positive valence vs. negative valence) of the sensory consumption.

Conclusions

Across four experiments, we find that individuals who are subjected to resource scarcity show a preference for high-intensity sensory consumption in the auditory, visual, and tactile domains. We also find that both negatively and positively valenced sensory stimuli have the same impact on resource-scarce individuals’ preference for HISC. Furthermore, we demonstrate that the sense of self-worth significantly mediates the effect of resource scarcity on HISC. Finally, we reveal that self-acceptance moderates the effect of resource scarcity on HISC preference.

2. During your revisions, please note that a simple title correction is required: The word "fill" should be replaced with "fills". Please ensure this is updated in the manuscript file and the online submission information.

Author Response: In the revised manuscript the word "fill" has been replaced with "fills" and updated in the manuscript file.

3. Please provide additional details regarding participant consent. In the Methods section, please ensure that you have specified (1) whether consent was informed and (2) what type you obtained (for instance, written or verbal). If your study included minors, state whether you obtained consent from parents or guardians. If the need for consent was waived by the ethics committee, please include this information.

Author Response: In the revised manuscript, we provide additional details regarding participant consent in the method of abstract and make it clear that all participants are adults.

Author Response: At the end of the revised manuscript, we added the captions of the supporting information file and any in-text citations to match accordingly. The added support information file titles are as follows:

S1 Appendix

S2 Figure

S3 Dataset

S4 Table

---

## [Decision Letter · Decision Letter 1]

25 Apr 2023

PONE-D-22-28511R1How high-intensity sensory consumption fills up resource scarcity: The boundary condition of self-acceptancePLOS ONE

Dear Dr. Haiyan Luo,

Thank you for submitting your manuscript to PLOS ONE. After careful consideration, we feel that it has merit but does not fully meet PLOS ONE’s publication criteria as it currently stands. Therefore, we invite you to submit a revised version of the manuscript that addresses the points raised during the review process. Please check the reviewer 1's comments and address the issues accordingly.

We look forward to receiving your revised manuscript.

Kind regards,

Abdullah Al Mamun, PhD

Academic Editor

PLOS ONE

Journal Requirements:

Reviewers' comments:

Reviewer's Responses to Questions

**Comments to the Author**

1. If the authors have adequately addressed your comments raised in a previous round of review and you feel that this manuscript is now acceptable for publication, you may indicate that here to bypass the “Comments to the Author” section, enter your conflict of interest statement in the “Confidential to Editor” section, and submit your "Accept" recommendation.

Reviewer #1: (No Response)

Reviewer #2: All comments have been addressed

2. Is the manuscript technically sound, and do the data support the conclusions?

Reviewer #1: No

Reviewer #2: Yes

3. Has the statistical analysis been performed appropriately and rigorously? 

Reviewer #1: No

Reviewer #2: Yes

4. Have the authors made all data underlying the findings in their manuscript fully available?

Reviewer #1: Yes

Reviewer #2: Yes

5. Is the manuscript presented in an intelligible fashion and written in standard English?

Reviewer #1: Yes

Reviewer #2: Yes

6. Review Comments to the Author

Reviewer #1: 1) Abstract: No. of participant were not disclosed and the DOE was quite simplified and more towards a vague arrangement.

2) Page 12: The no. of participant was only mentioned at this section (182 participant) and this should be clearly worth mentioning earlier for a better understanding.

3) The specific statistical analysis in the method section was quite unclear and need more info on that.

4) The ultimatum comment: The DOE need to be clearly explained and visualized (e.g. Table and Figures) so that the reviewer could understanding at the fullest of the outcome of this studies. Not to mention it is quite excruciating lengthy with unclear statistical approaches.

Reviewer #2: The paper has improved a lot by the revisions. No further actions required.

The paper has improved a lot by the revisions. No further actions required.

7. PLOS authors have the option to publish the peer review history of their article (what does this mean?). If published, this will include your full peer review and any attached files.

Reviewer #1: **Yes: **Saiful Irwan Zubairi

Reviewer #2: **Yes: **Anne Berthold

---

## [Author Response · Author response to Decision Letter 1]

30 Apr 2023

Response to Reviewer 1

Reviewer#1, Concern # 1: Abstract: No. of participant were not disclosed and the DOE was quite simplified and more towards a vague arrangement.

Author Response: First, we fully disclosed the participants in the Abstract section in revised manuscript. Second, we elaborate the DOE of Abstract section (including participants, variable measurement methods, statistical analysis methods, experimental results) so that it is clearly presented to the reader. Third, we reinforced the DOE in the first paragraph of each study.

Author action: 

(1) In the methods section of the Abstract, we update the manuscript as follows:

We used different methods including one-way ANOVA, linear regression, mediating effect, and moderating effect to test the four hypotheses. Four experiments in the study were conducted from May 2022 and August 2022 and involved undergraduates from a university and volunteers recruited online. All participants are adults and verbally agree to participate voluntarily. Study 1a (N = 96 (male 47, female 49), participants from a business school in China) measured resource scarcity in the laboratory experiments and verified the effect of resource scarcity on consumer HISC preference by using linear regression (H1). Study 1b (N = 191 (male 98, female 93), students and teachers from a university in China) measured resource scarcity in the laboratory experiments and manipulated positively and negatively valenced experiences. Using the PROCESS SPSS Mode l, we verified that negatively valenced stimuli also lead to higher levels of arousal, which in turn restores the self-discrepancy caused by resource scarcity (H2). Study 2 (an online experiment, N = 182 (male 91, female 91), participants from China) manipulated resource scarcity in color sensory stimulant context, replicating the preliminary effect and examined the mediating effect of the self-worth by using the PROCESS SPSS Mode 4 (H3). Study 3 (an online experiment, N = 251 (male 125, female 126), participants from China) manipulated resource scarcity and self-acceptance in tactile sensory experience, and tested the moderating effect of self-acceptance by using the PROCESS SPSS Mode 8 (H4). 

(2) We reinforced the DOE in the first paragraph of each study as follows:

Study 1a was designed to test H1 by using linear regression method to study the effect of resource scarcity on consumers’ audio consumption choices. Study 1a was conducted in a laboratory and recruited 96 participants from a business school in China. We measured participants' perceived resource scarcity in laboratory experiments.

An important goal that study 1b was designed to test H2 by using PROCESS SPSS Mode l method to examine what effect the valence of HISC may have on the observed impacts. Study 1a was conducted in a laboratory and recruited 191 students and teachers from a business school in China. We manipulated participants' positively and negatively valenced experiences.

The design of Study 2 is to test H3 by using PROCESS SPSS Mode l method to document the mediating role of self-worth between resource scarcity and HISC in the visual domain. Study 2 was conducted on a research service online platform (www.credamo.com) called Credamo and recruited 182 voluntary participants. We manipulated participants' resource scarcity in color sensory stimulant context.

The design purpose of study 3 is to test H4 by using PROCESS SPSS Mode 8 to investigate the moderating effect of self-acceptance on resource-scarce individuals’ preference for HISC in the tactile domain. Study 3 was conducted on a research service online platform (www.credamo.com) called Credamo and recruited 251 voluntary participants. We manipulated participants' resource scarcity and self-acceptance in tactile sensory experience.

Reviewer#1, Concern # 2: Page 12: The no. of participant was only mentioned at this section (182 participant) and this should be clearly worth mentioning earlier for a better understanding. 

Author Response: We fully understand the reviewer's comments. As in Reply to Concern # 1, we advance participants to the first paragraph of each study (Study 1a, Study 1b, Study 2, Study 3) because this should be clearly worth mentioning earlier for a better understanding. 

Author action: we advance participants to the first paragraph of each study (Study 1a, Study 1b, Study 2, Study 3) as follows:

Study 1a was designed to test H1 by using linear regression method to study the effect of resource scarcity on consumers’ audio consumption choices. Study 1a was conducted in a laboratory and recruited 96 participants from a business school in China. We measured participants' perceived resource scarcity in laboratory experiments.

An important goal that study 1b was designed to test H2 by using PROCESS SPSS Mode l method to examine what effect the valence of HISC may have on the observed impacts. Study 1a was conducted in a laboratory and recruited 191 students and teachers from a business school in China. We manipulated participants' positively and negatively valenced experiences.

The design of Study 2 is to test H3 by using PROCESS SPSS Mode l method to document the mediating role of self-worth between resource scarcity and HISC in the visual domain. Study 2 was conducted on a research service online platform (www.credamo.com) called Credamo and recruited 182 voluntary participants. We manipulated participants' resource scarcity in color sensory stimulant context.

The design purpose of study 3 is to test H4 by using PROCESS SPSS Mode 8 to investigate the moderating effect of self-acceptance on resource-scarce individuals’ preference for HISC in the tactile domain. Study 3 was conducted on a research service online platform (www.credamo.com) called Credamo and recruited 251 voluntary participants. We manipulated participants' resource scarcity and self-acceptance in tactile sensory experience.

Reviewer#1, Concern # 3: The specific statistical analysis in the method section was quite unclear and need more info on that.

Author Response: We have revised the method part of the abstract, in which we separately added the participants, the variable measurement method, and the specific statistical analysis method of each experiment. The specific statistical analysis methods used in experiment 1a, experiment 1b, experiment 2 and experiment 3 are linear regression, PROCESS SPSS Mode l, PROCESS SPSS Mode 4, and PROCESS SPSS Mode 8, respectively.

Author action: we added the specific statistical analysis in the method section as follows:

We used different methods including one-way ANOVA, linear regression, mediating effect, and moderating effect to test the four hypotheses. Four experiments in the study were conducted from May 2022 and August 2022 and involved undergraduates from a university and volunteers recruited online. All participants are adults and verbally agree to participate voluntarily. Study 1a (N = 96 (male 47, female 49), participants from a business school in China) measured resource scarcity in the laboratory experiments and verified the effect of resource scarcity on consumer HISC preference by using linear regression (H1). Study 1b (N = 191 (male 98, female 93), students and teachers from a university in China) measured resource scarcity in the laboratory experiments and manipulated positively and negatively valenced experiences. Using the PROCESS SPSS Mode l, we verified that negatively valenced stimuli also lead to higher levels of arousal, which in turn restores the self-discrepancy caused by resource scarcity (H2). Study 2 (an online experiment, N = 182 (male 91, female 91), participants from China) manipulated resource scarcity in color sensory stimulant context, replicating the preliminary effect and examined the mediating effect of the self-worth by using the PROCESS SPSS Mode 4 (H3). Study 3 (an online experiment, N = 251 (male 125, female 126), participants from China) manipulated resource scarcity and self-acceptance in tactile sensory experience, and tested the moderating effect of self-acceptance by using the PROCESS SPSS Mode 8 (H4). 

Reviewer#1, Concern # 4: The ultimatum comment: The DOE need to be clearly explained and visualized (e.g. Table and Figures) so that the reviewer could understanding at the fullest of the outcome of this studies. Not to mention it is quite excruciating lengthy with unclear statistical approaches.

Author Response: First, we revised the method section of the Abstract, which clearly explains the design of experiment for each study including participants, variable measurement methods, statistical analysis methods. Second, we revised the first paragraph of study 1a, study 1b, study2, and study3, which clearly explains the design of experiment including the purpose of the experiment, participants, variable measurement methods, statistical analysis methods 

Third, each of our studies has visual research results, which are included in the result section. Outcome of this studies are visualized in 10 Figures, which uploaded separately according to journal requirements and are not included in the manuscript. In order to enable reviewer to understand at the fullest of the outcome of this studies, we show the figures below.

Author action: 

First, we update the method section of the Abstract as follows:

We used different methods including one-way ANOVA, linear regression, mediating effect, and moderating effect to test the four hypotheses. Four experiments in the study were conducted from May 2022 and August 2022 and involved undergraduates from a university and volunteers recruited online. All participants are adults and verbally agree to participate voluntarily. Study 1a (N = 96 (male 47, female 49), participants from a business school in China) measured resource scarcity in the laboratory experiments and verified the effect of resource scarcity on consumer HISC preference by using linear regression (H1). Study 1b (N = 191 (male 98, female 93), students and teachers from a university in China) measured resource scarcity in the laboratory experiments and manipulated positively and negatively valenced experiences. Using the PROCESS SPSS Mode l, we verified that negatively valenced stimuli also lead to higher levels of arousal, which in turn restores the self-discrepancy caused by resource scarcity (H2). Study 2 (an online experiment, N = 182 (male 91, female 91), participants from China) manipulated resource scarcity in color sensory stimulant context, replicating the preliminary effect and examined the mediating effect of the self-worth by using the PROCESS SPSS Mode 4 (H3). Study 3 (an online experiment, N = 251 (male 125, female 126), participants from China) manipulated resource scarcity and self-acceptance in tactile sensory experience, and tested the moderating effect of self-acceptance by using the PROCESS SPSS Mode 8 (H4).

Second, we update the first paragraph of study 1a, study 1b, study2, and study3 as follows:

Study 1a was designed to test H1 by using linear regression method to study the effect of resource scarcity on consumers’ audio consumption choices. Study 1a was conducted in a laboratory and recruited 96 participants from a business school in China. We measured participants' perceived resource scarcity in laboratory experiments. We examine the propensity for HISC by participants’ attitude toward an intense (high decibel) sound and their preferred volume on continuous audio in the same audio clips ranging from less intense to high intense sound (low decibel to high decibel). The audio sound we used in study 1a is music clips without lyrics.

An important goal that study 1b was designed to test H2 by using PROCESS SPSS Mode l method to examine what effect the valence of HISC may have on the observed impacts. Study 1a was conducted in a laboratory and recruited 96 students and teachers from a business school in China. We manipulated participants' positively and negatively valenced experiences. Unlike study 1a, which used only positively valenced sensory stimuli, study 1b included negatively valenced stimuli. We expect that individuals subjected to resource scarcity prefer negatively valenced stimuli just as they prefer positively valenced stimuli, since negatively valenced stimuli also lead to higher levels of arousal, which in turn restores the self-discrepancy caused by resource scarcity.

In studies 1a and 1b, we demonstrate that individuals subjected to resource scarcity exhibit a propensity for HISC in the auditory domains. We speculate that resource scarcity would reduce an individual’s self-worth and that HISC itself, like other affirming activities, serves as a restorative mechanism of self-worth [105] (Steele 1988; Townsend, Sood 2012). The design of Study 2 is to test H3 by using PROCESS SPSS Mode l method to document the mediating role of self-worth between resource scarcity and HISC in the visual domain. Study 2 was conducted on a research service online platform (www.credamo.com) called Credamo and recruited 182 voluntary participants. We manipulated participants' resource scarcity in color sensory stimulant context. We hypothesize that resource scarcity reduces the individual’s self-worth, which in turn promotes an individual’s HISC preference and ultimately has a restorative effect on self-worth. Unlike studies 1a and 1b, in study 2 we examine the propensity for HISC by the attitude of the participants toward intense color (highly saturated) and by their preferred choice of color on a continuum of shades ranging from less intense (less saturated) to more intense colors (more saturated) in the same hue family.

The design purpose of study 3 is to test H4 by using PROCESS SPSS Mode 8 to investigate the moderating effect of self-acceptance on resource-scarce individuals’ preference for HISC in the tactile domain. Study 3 was conducted on a research service online platform (www.credamo.com) called Credamo and recruited 251 voluntary participants. We manipulated participants' resource scarcity and self-acceptance in tactile sensory experience. Study 3 shows that individuals subjected to resource scarcity prefer HISC in the visual domain when they are not allowed additional opportunities to restore their self-worth, which is mediated by self-worth. First, Study 3 is designed to investigate the robustness of the effects observed in previous studies by testing whether resource scarcity impacts the preference for HISC in the tactile domain. Second, Study 3 will expand on previous studies by testing whether self-acceptance attenuates the mediating effect of Study 3.

Third, Outcome of this studies are visualized in 10 Figures, which are shown below.

Fig. 1. Conceptual framework.

Fig. 2 Moderating effect of valence (negative vs. positive) on the resource scarcity and HISC

Fig. 3. Orange color swatches of high- and low- intensity visual stimuli

Fig. 4 Spectrum of low-high intensity color

Fig. 5. The mediating effect of sense of the self-worth.

Fig. 6 Moderating effect of self-acceptance on the resource scarcity and self-worth

Fig. 7 Moderating effect of self-acceptance on the resource scarcity and HISC

Response to editor and reviewers

Dear Editor,

Thank you for allowing a resubmission of our manuscript, with an opportunity to address the reviewer comments.

We have uploaded on the system (a) our point-by-point responses to comments (responses to the reviewer and editor), (b) the revised manuscript with the changes shown, and (c) the cleanly updated manuscript with no changes shown if the submission system is accessible.

In this revised manuscript, we have removed three references that may have been retracted and replace them with relevant current references. As authors, we don't have the ability to find out whether the literature has been retracted, but the literature we cite has been downloaded and read. In addition, we have uploaded the data as a supplement to the submission system.

Best regards

Liangjun Peng, Yuxin Peng, Haiyan Luo, Yeying Den

---

## [Editor Report · Decision Letter 2]

3 May 2023

How high-intensity sensory consumption fills up resource scarcity: The boundary condition of self-acceptance

PONE-D-22-28511R2

Dear Dr.Haiyan Luo,

We’re pleased to inform you that your manuscript has been judged scientifically suitable for publication and will be formally accepted for publication once it meets all outstanding technical requirements.

Kind regards,

Abdullah Al Mamun, PhD

Academic Editor

PLOS ONE

Additional Editor Comments (optional):

Congratulations
---

## [Editor Report · Acceptance letter]

18 May 2023

PONE-D-22-28511R2 

How high-intensity sensory consumption fills up resource scarcity: The boundary condition of self-acceptance 

Dear Dr. Luo:

I'm pleased to inform you that your manuscript has been deemed suitable for publication in PLOS ONE. Congratulations! Your manuscript is now with our production department. 

Kind regards, 

on behalf of

Assoc. Prof. Dr. Abdullah Al Mamun 

Academic Editor

PLOS ONE